# Predictors of Excess Return in a Green Energy Equity Portfolio: Market Risk, Market Return, Value-at-Risk and or Expected Shortfall?

**Rebecca Abraham** [1,*] **, Hani El-Chaarani** [2] **and Zhi Tao** [3]

1    Huizenga College of Business, Nova Southeastern University, 3301 College Ave., Fort Lauderdale, FL 33314, USA
2    College of Business Administration, Beirut Arab University, P.O. Box 1150-20, Riad El Solh 11072809, Lebanon; h.shaarani@bau.edu.lb or hanichaarani@hotmail.com
3    College of Business and Public Policy, University of Alaska-Anchorage, 3211 Providence Drive, Anchorage, AK 99508, USA; ztao@alaska.edu
*    Correspondence: abraham@nova.edu

**Abstract:** The rapid growth of electric vehicles, solar roofs, and wind power suggests that the potential growth in green equity investments is an emerging trend. Accordingly, this study measured the predictors of excess equity returns in a portfolio of global green energy producers, from 2010 to 2019. Fixed-effects panel data regressions of daily returns, followed by quantile regressions, were performed. There was some support for the explanation of green equity returns by market returns and market risk (beta), as indicated by the single-factor Capital Asset Pricing Model (CAPM), and the multifactor Fama–French Three-Factor and Fama–French Five-Factor Models. The most significant predictors of green equity returns were Value-at-Risk at a 95% confidence level, and Value-at-Risk at a 99% confidence level. Expected Shortfall was another extreme risk value measure. The importance of extreme value measures suggests the presence of fat-tailed leptokurtic distributions, whereby excess returns were explained by the risk of loss given adverse conditions, primarily at 95% confidence. We conclude that the proliferation of small firms and new entrants in the renewable energy sector has led to the explanation of returns by extreme values of risk.

**Keywords:** green equities; Capital Asset Pricing Model; Fama–French Three-Factor Model; Fama–French Five-Factor Model; Value-at-Risk; Expected Shortfall

## 1. Introduction

Spurred on by fears of climate change, there has been a surge in fossil fuel substitutes, such as electric vehicles, solar roofs, and the use of wind energy for home energy usage. In other words, the global marketplace is adapting to a future with a limited carbon footprint. As an example, the >$1000 stock price for Tesla in 2021, from low double-digit values in the prior three years, portends the potential for explosive growth in the green energy sector. Likewise, Rivian Automotive, another electric vehicle producer, has experienced price surges in the pre-IPO period. It follows that investments in green securities may become increasingly attractive as public and private funds flow to producers of fossil fuel substitutes. Environmentally conscious investors may wish to capitalize on this growth by purchasing mutual funds or stocks of renewable energy producers. Yet, the academic literature in this area is sparse.

Early studies of green portfolio performance generated conflicting results. Brzeszczynski and McIntosh's (2014) comparison of socially responsible securities of the 100 leading sustainable corporations with market indexes yielded higher returns for the entire 2000–2010 period. Only market returns could explain socially responsible portfolio returns, using the Fama–French model. Likewise, Cai and He (2014) observed higher returns for

securities of environmentally responsible firms in the fourth through the seventh year after the screening year of the 1992–2011 period. In contrast, Ibikunle and Steffen's (2017) green portfolio significantly underperformed conventional black energy portfolios, due to their greater concentration of small cap and growth stocks. Mixed results were obtained from Lesser et al.'s (2014) comparison of green investing, with socially responsible investing with green equities having higher returns than socially responsible equities from 2003 to 2007, with underperformance from 2008 to 2012.

However, these early studies were restricted to the market model, theorizing that security returns were dependent upon co-movements of green equity portfolios with the market portfolio, along with balance sheet quantities such as income and profitability. Their study did not capture the effects of data points found in the tails of stock price distributions. We attempt to fill this research gap by measuring the effects of market returns and beta assuming the market model, followed by predictors of excess return by idiosyncratic risk measures, typically found in the lower tail of the distributions of green equities. We employ three idiosyncratic tail risk measures, the Value-at-Risk at 95% confidence, the Value-at-Risk at 99% confidence, and the Expected Shortfall, as we feel that identifying the specific form of idiosyncratic risk contributes to knowledge.

We advance knowledge in three ways. First, we offer a comprehensive examination of the predictors of excess returns in green securities, by using three market models, and three idiosyncratic risk models. This is in contrast to the Lesser et al. (2014) study, which confined itself to the market model, two market-related models, and balance sheet predictors. Second, we cover a time period that has not been considered in the literature, i.e., 2010–2019, which uses more contemporary data than the earlier studies, which do not have samples beyond 2012. The entry of new firms, and the abnormal growth of others, such as Tesla, can only be captured by a more current sample. Third, we add to the literature that shifts the focus of discussion about excess returns from the market model, with predictable measures of risk, such as beta, to the largely unknown tail risk, that has achieved prominence through extremely high-risk investments, such as cryptocurrencies. Two studies are noteworthy in this regard. Bouri et al. (2017) observed that crude oil and gild were weak buffers to extreme down movements in energy stock indices, i.e., down movements captured by tail risk. Saeed et al. (2021) showed that returns of crude energy securities and green energy securities showed a high degree of connectedness in the tails, though not at the mean. We add to these findings by evaluating green portfolio performance using both market models, and extreme value measures, to provide a single study that includes both traditional market-based measures and tail risk measures to explain excess returns in green energy securities.

The remainder of this paper is organized as follows. Section 2 describes the background literature and develops hypotheses. Section 3 describes Materials and Methods. Section 4 displays Results, while Section 5 delineates Discussions.

## 2. Review of Selected Literature and Hypotheses Development

### 2.1. The Capital Asset Pricing Model

Modern portfolio theory originated with Markowitz's (1952) postulate that portfolio selection was optimized by selecting portfolios on an investment frontier with the highest expected return and lowest variance risk. A specific one-factor portfolio valuation model that defined the relationship between security return and risk was the Capital Asset Pricing Model (CAPM) (Lintner 1965; Sharpe 1964). The expected return on a security is the sum of the lowest possible return, the risk-free rate, and the excess of the security return above the market return. The coefficient of the excess of the security return above the market return is the beta coefficient, a measure of the security's sensitivity to market risk.

$$R_j = R_f + b_j \left( R_m - R_f \right) \tag{1}$$

where

$R_j$ = the expected return on a security,
$R_f$ = the risk-free rate,
$b_j$ = beta coefficient, and
$R_m$ = market return.

In Equation (1), the excess return is $(R_j - R_f)$.
Rearranging Equation (1) yields Equation (2), which measures excess return,

$$(R_j - R_f) = b_j \left( R_m - R_f \right) \tag{2}$$

Chen's (2021) review presented multiple applications of CAPM. Regulatory CAPM envisioned beta as the risk of firms involved in the transport of water and wastewater. The Sharpe ratio of risk per unit of return viewed beta as the sensitivity of an index fund to portfolio returns. Beta, thus, measures the ability of a portfolio manager to generate returns equal to the broad market. As Shalit and Yitzakhi (2003) indicated, CAPM's assumption of identical risk-return proclivities among investors may be challenged by the heterogeneity among investors. All risk-averse investors are unlikely to hold similar portfolios, so that some investors may hold portfolios that are different from the mainstream risky portfolio and rely on cash or borrowing at the risk-free rate to manage risk (Tobin 1958).

Applying CAPM to green energy equities, the excess return on green energy portfolios depends on the returns on the broad market, and the risk of the broad market. The underlying logic is that a security has risk. Returns are commensurate with the level of risk, so that its return increases as its risk rises above the risk-free rate. The relevant risk is the non-diversifiable risk, beta, which is attributed to the sensitivity if the security's return to the return on the broad market. However, negative betas are possible. Such a portfolio offsets declines in the broad market without short-selling (Chen 2021). A green energy equity portfolio with negative betas earns excess returns during periods of broad market declines. Possibly the excess returns are being earned by securities of wind, solar, or electric vehicle firms that are so new that their movements have not been integrated into the broad market. Among Chen's (2021) applications of CAPM, heterogeneity of investor predispositions is relevant to returns on green energy equities. Investors in green energy equities may seek returns in risky green equity startups, so that their investment will have greater investment in the risky portfolio, and less investment in cash, or borrowing at the risk-free rate. We state Hypothesis 1 as follows,

**Hypothesis 1.** *The excess return on a portfolio of green energy equities is a function of market return, $R_m$, and its sensitivity to (1) market risk, beta, positive or negative, or (2) no relation to beta, due to differences between investors in green energy equities, and investors in the broad market. This was true for a portfolio of global green energy equities from 2010 to 2019.*

### 2.2. The Fama–French Multifactor Models

CAPM has anomalies. It makes the theoretical assumption of the existence of a hypothetical portfolio of all of the stocks in the market. Roll (1977) showed that the returns on such a portfolio are not measurable, as it does not exist. Further, the sensitivity to market risk alone is insufficient to capture the risk inherent in a security. This is borne out by empirical examinations of the single-factor CAPM, wherein the positive relationship between stock returns and market returns assumed by CAPM was only found in a few cases (see Benz 1981), for a review). Investors perceive other forms of risk, than market risk, and therefore returns must compensate investors for accepting these risks. Fama and French (1996) added firm characteristics, such as size, and book-to-market, as predictors of expected return to the Capital Asset Pricing Model. Small firms typically are riskier, so that their returns may need to be higher than those of large firms. Benz (1981) found that CAPM's predictive capacity increased with the addition of a variable that accounted for size. Book-to-market differentiates between high book value and high market value securities. High book value securities are typically high-performance firms that are less

well known, so do not garner attention from analysts, the press, or the public. In contrast, high market value stocks dominate the news, but may underperform as all of the available information about their performance has been incorporated into prices. A series of studies have provided empirical support for the superior predictive power of the three-factor Fama–French model over CAPM (Benz 1981; Fama and French 2012; Griffin 2002).

In the context of green energy equities, large, established firms are more likely to have returns that depend on the broad market. The entry of small producers of solar energy, wind energy, and electric vehicles suggests that their returns will be different from their larger counterparts. Some, such as Rivian Automotive, will grow rapidly, creating value, or with high book to market value. Others may enter the renewables sector as glamour stocks, fail to fulfill their initial promise, and exit the market, such as Bright Autonotive, Coda, and Aptera. Therefore, factors must be added to CAPM, to account for size, and book-to-market differentials. We state the second hypothesis as follows.

**Hypothesis 2.** *The excess return on a portfolio of green energy equities is a function of market return, $R_m$, and its sensitivity to market risk, beta, along with sensitivity to size differentials and book-to-market differentials. This was true for a portfolio of global green energy equities from 2010 to 2019.*

The Fama–French five-factor model included factors for investment and profitability. The underlying logic was based on the dividend discount valuation model, which viewed stock returns as the discounted value of future dividends.. Higher projected profitability results in higher projected cash flows, and in turn, higher stock returns. Industries with high profit margins, such as the pharmaceutical industry, have lower risk upon confronting economic downturns, than consumer cyclical industries. For example, establishments in leisure and hospitality, with thin profit margins, experienced excessive disruptions during the lockdowns in 2020, as opposed to producers of pharmaceuticals, during the same period of time. Conversely, higher levels of fixed investment reduce future cash flows, resulting in expectations of lower stock returns (Dolinar et al. 2020). Firms in the oil and gas industry that purchase expensive drilling equipment, while investing in risky oil exploration projects, have high fixed investment. Their cash flows may be adversely affected, resulting in lower future stock returns. Securities can differ based on investment and profitability (Fama and French 2014). Given differential levels of investment and profitability of green energy producers, it is appropriate to include these factors as predictors of excess return on green energy securities.

**Hypothesis 3.** *The excess return on a portfolio of green energy stocks is a function of market return, $R_m$, and its sensitivity to market risk, beta, along with sensitivity to size differentials, book-to-market differentials, fixed investment differentials, and profitability differentials. This was true for a portfolio of global green energy equities from 2010 to 2019.*

*2.3. Extreme Value Theory*

Extreme value theory maintains that large fluctuations in security returns may arise due to the effects of abrupt shifts in economic policy, crises, wars, economic shocks (Jammazi and Nguyen 2017), such as inflation, and the cessation of business from pandemics. CAPM's assumption of normality of stock returns is violated in such situations with nonlinearity, skewed distributions, heavy tails (numerous returns have very high or very low values, clustering in the tails, rather than towards the center of the distribution), and leptokurtic distributions which are flat due to the heavy tails, with few observations at the mean. Jammazi and Nguyen's (2017) review concluded that omitting these irregularities from financial time series would lead to erroneous conclusions. Studies that have attempted to explain variations in stock returns due to risk contained in fat tails have frequently demonstrated superior explanatory power than market models (Aziz and Ansari 2017). Aziz and Ansari (2017) posited that idiosyncratic volatility (as existed in the tails) was

positively related to future stock returns. More importantly, they observed that insignificant relationships at the center of a distribution may become significant in extreme quantiles at the edge of the conditional distribution. Huang et al. (2012) created an extreme downside risk (EDR) proxy in the left tail of an extreme value distribution, finding that a significant extreme downside risk premium existed after controlling for market size, value, momentum, and liquidity. They observed that high EDR stocks had high idiosyncratic risk, large downside beta, lower co-skewness and co-kurtosis, and high bankruptcy risk. In short, idiosyncratically volatile stocks, and high EDR stocks, were capable of exhibiting risk-return characteristics that were atypical from the broad market. We will evaluate the measures of extreme value risk, Value-at-Risk and Expected Shortfall.

2.3.1. Value-at-Risk

Value-at-Risk measures the loss potential of a security. For example, a 5% VAR of -2.3% indicates that we can be 95% confident that the security return will not be less than 2.3%. This definition includes the minimum loss potential (2.3%), along with the probability of exceeding the loss (95%). As a percentage of the portfolio, assuming a portfolio of $20,000, the VAR is = 0.023*20,000 = $460. This is the minimum amount of capital to offset the risk of loss.

It has been used to explain security returns in volatile markets, with fat-tailed distributions of security returns. Value-at-Risk may be computed as follows.

$$VAR\alpha = Z\alpha\sigma \tag{3}$$

where

$\alpha$ = 95 or 99, the VAR percentile,
$Z$ = Z score of the VAR percentile, and
$\sigma$ = standard deviation of security returns.

Emerging markets typically exhibit fat tails of security returns. Chen and Giles (2014) found that security returns on the Taiwan stock exchange were explained by Value-at-Risk, as did Aziz and Ansari (2017) for the Indian stock exchange. Iqbal and Azher (2014) observed a significant risk premium upon using Value-at-Risk as a measure of risk for a sample of stocks traded on the Pakistan Stock Exchange.

Could green energy securities have tail risk that could be effectively measured by Value-at-Risk? The industry is in its initial phase of the product life cycle. Many new entrants have entered the market for solar power, wind energy, and electric vehicles. In solar energy, as prices have declined with the reduction in cost of production, many of the new entrants have experienced cost overruns, leading to the shrinking of profit margins. Declines in state subsidies for home and business solar installations have further shrunk profit margins, so that only the most financially sound firms have survived. For wind energy, individual homeowners have been selling wind power to utility companies. Numerous conflicts have emerged, with disputes over the amount of compensation. Further, different levels of wind power in different parts of the US have resulted in inequities in benefits from wind energy sales, leading to more disputes. Uncertainty about wind power prices abounds. The electric vehicle market is ripe for consolidation, with a few firms having survived labor disputes, parts supply chain disruptions, and lack of battery power availability. Overall market growth is limited by the scarcity of charging stations, and uncertainty about future government subsidies encouraging electric vehicle purchases. Perhaps most importantly, the market may fail to grow in the near future, as the high prices of electric vehicles reduce purchases from cash-constrained consumers. In essence, uncertainties in these industries suggest that green energy securities may experience tail risk, which could be measured by Value-at-Risk.

**Hypothesis 4a.** *The excess return on a portfolio of green energy stocks is a function of tail risk as measured by Value-at-Risk at 95% confidence. This was true for a portfolio of global green energy equities from 2010 to 2019.*

**Hypothesis 4b.** *The excess return on a portfolio of green energy stocks is a function of tail risk as measured by Value-at-Risk at 99% confidence. This was true for a portfolio of global green energy equities from 2010 to 2019.*

2.3.2. Expected Shortfall

While Value-at-Risk is a parametric measure, assuming normally distributed stock returns, accompanied by excessive skewness, or excessive kurtosis, Expected Shortfall is a non-parametric measure of tail risk. While Value-at-Risk measures the percentile of a loss distribution, Expected Shortfall measures the average of expected losses in the percentile defined by Value-at-Risk. For a 4% VAR of $500, there is a 96% probability that a loss of more than $500 will not occur. Within the 4% VAR percentile, average losses may be $560. This figure is the Expected Shortfall. Sarykalin et al. (2008) set forth that Expected Shortfall is the more comprehensive measure of risk in the tails of an extreme value distribution, as it presents the losses beyond the Value-at-Risk percentile, while Value-at-Risk does not measure them. Expected Shortfall has robust mathematical properties with continuous convex functions, in contrast with Value-at-Risk, which may have discontinuous functions. The Expected Shortfall is measured as follows.

$$Expected\ Shortfall = (1/(1-c)) \int -1\ to\ VAR\ xp(x)dx \tag{4}$$

$c$ = Value-at-Risk breakpoint of the distribution, and
$p(x)$ = probability density function of obtaining returns with value $x$.

Given the high levels of uncertainty associated with prices of solar installations, compensations for wind power, and intense competition among electric vehicle producers, along with macroeconomic uncertainties including regulation, inflation, and legislative vacillations on the provision of subsidies for electric vehicles, investors may expect a range of expected losses upon making green energy equity purchases. Expected Shortfall is the worst average expected loss of these expected losses, which may explain excess returns. We state the fifth hypothesis as follows.

**Hypothesis 5.** *The excess return on a portfolio of green energy stocks is a function of tail risk as measured by Expected Shortfall at 95% confidence. This was true for a portfolio of global green energy equities from 2010 to 2019.*

## 3. Materials and Methods

### 3.1. Sample and Data Characteristics

A sample of 122 publicly traded firms engaged in the global production of wind energy, solar energy, electric vehicle production, nuclear energy to combat global warming, clean coal, ethanol, geothermal energy, and waste management was selected by perusing online and print news sources. Some of the firms were electric power companies with renewable energy divisions, and a sustained commitment to renewables, although they were currently engaged in traditional forms of energy production. Others were energy funds. Appendix A shows a listing of the stocks. The sample size may appear small; however, we justify it as we screened out firms with incomplete Fama–French risk measures, and firms that failed to demonstrate an ESG commitment, as indicated by their mission statements. It was important to us that there be continuity of production of renewable energy, or electric vehicles, i.e., that the firms did not view ESG as a fad, to be tried and discarded. We justify the time period chosen, as data are complete for the 2010–2019 period, and the sample captures the evolution of renewable energy production from being part of a large firm's production (and hence driven by the market model), to small firm production (and hence, represented by extreme value distributions). Data were collected from 2010 to 2019, with daily returns, beta, market capitalizations, book-to-market and assets values shown in Table 1, Panel A, using the global COMPUSTAT database. The sample period was chosen as it was expected that the market model would dominate in early years as indicated

in the literature (Cai and He 2014; Lesser et al. 2014), while giving way to idiosyncratic risk in the tails as explaining excess returns in later years. We wished to have a dataset that captures the transition from excess returns being explained by the market model, to excess returns being explained by idiosyncratic risk. The period prior to 2010 was not included, as it is likely that the market model dominated in these years. The large standard deviations of market capitalization and asset size attest to the presence of both small and large producers of green energy in the sample. Likewise, the wide range of book-to-market values suggests the inclusion of less-known value stocks, and well-known glamour stocks. Table 1, Panel B shows the maximum and minimum values of each variable. Some countercyclical movement of security prices may be observed, with negative beta values, as security prices varied in opposition to the direction of movement of the market. Panel C displays the correlation matrix of the one dependent variable, and six independent variables. Should variables be tested for stationarity? We posit that since our analysis is confined to ultra-short holding periods of 1 year, and since stationarity assumes stable long-run relationships, with short-run variation, our study captures the short-term return differences, while not measuring long-term return influences, stationarity may not need to be measured.

**Table 1.** Descriptive statistics. Panel A: Means and Standard Deviations of Dependent and Independent Variables. Means are listed first in each cell, followed by standard deviations in parentheses. Panel B: Maxima and Minima of Dependent and Independent Variables. Maximum values of variables are listed first in each cell, followed by minimum values. Panel C: Correlation Matrix of Key Variables, 2019.

| | | | | | Panel A | | | | | |
|---|---|---|---|---|---|---|---|---|---|---|
| Variable | 2010 | 2011 | 2012 | 2013 | 2014 | 2015 | 2016 | 2017 | 2018 | 2019 |
| Excess Return | −0.03 (0.07) | 0.03 (4.9) | −0.008 (1.1) | −0.003 (0.41) | 0.01 (0.002) | 0.06 (5.16) | −0.007 (0.59) | 0.003 (0.55) | 0.007 (1.6) | 0.009 (0.73) |
| Market Return | $3 \times 10^3$ ($9 \times 10^{-5}$) | $1 \times 10^{-4}$ (0.01) | $5 \times 10^{-4}$ ($7 \times 10^3$) | $8.1 \times 10^{-3}$ (0.002) | $4 \times 10^{-4}$ (1.9) | $1.8 \times 10^{-5}$ (0.007) | $3.9 \times 10^{-4}$ (0.01) | $7 \times 10^{-4}$ (.01) | −0.001 (0.01) | $9.8 \times 10^{-4}$ (0.007) |
| Beta | 0.72 (0.67) | 1.27 (1.0) | 1.28 (1.0) | 1.15 (1.05) | 1.16 (1.0) | 1.11 (1.06) | 1.27 (1.01) | 1.26 (0.97) | 1.24 (0.97) | 1.26 (1.0) |
| Market Capitalization (Log) | 5.53 (4.23) | 19 (5.29) | 18.5 (5.9) | 19 (6.0) | 20 (4.5) | 3.8 (3.02) | 19.7 (5.2) | 19.7 (4.8) | 20.5 (3.3) | 4.1 (2.9) |
| Book-to-Market | $3.5 \times 10^{-4}$ (0.001) | 1071 (4283) | 1045 (413) | 5360 (648) | 2723 (44) | $4.03 \times 10^{-4}$ (0.002) | 463 (911) | 902 (257) | 1077 (90) | $4.2 \times 10^{-4}$ (0.001) |
| EBIT (millions) | 167.7 (278) | 308.7 (1102) | 289 (10) | 308 (911) | 292.1 (51) | 73.88 (212.8) | 258 (197) | 223 (195) | 243 (51) | 33.41 (568) |
| Assets (millions) | | 24,797 (1100) | 2534 (11) | 1124 (18) | 6917 (30) | | 10,968 (222) | 2199 (438) | 2550 (70) | 250.3 (4.09) |
| | | | | | Panel B | | | | | |
| Variable | 2010 | 2011 | 2012 | 2013 | 2014 | 2015 | 2016 | 2017 | 2018 | 2019 |
| Excess Return | 42.27 −2.05 | 587 −2.11 | 127 −2.08 | 0.04 −0.03 | 202 −2.11 | 0.04 −0.03 | 55.65 −2.15 | 63.5 −2.07 | 210.21 −2.14 | 62.57 −2.06 |
| Market Return | −0.04 −0.03 | 0.04 −0.06 | 0.02 −0.02 | 33.4 −2.1 | 0.04 −0.06 | 595 −2.1 | 0.02 −0.03 | 0.07 −0.05 | 0.11 −0.09 | 0.03 −0.02 |
| Beta | 1.84 0 | 5.33 −0.27 | 5.33 −0.27 | 5.33 −0.27 | 5.33 −0.27 | 5.33 −1.19 | 5.33 −1.19 | 5.33 −1.19 | 5.33 −1.19 | 5.33 −0.83 |
| Market Cap | 14.56 0 | 26 0 | 26 0 | 26.36 0 | 26.29 0 | 8.49 0 | 25 0 | 26.27 −0.85 | 25.49 2.10 | 8.07 0 |
| Book-to-Market | $5.2 \times 10^{-3}$ $−4.7 \times 10^{-3}$ | 28,590 0 | 27,134 −318 | 79,147 0 | 24.91 −15.5 | 0.21 0 | 3636 0 | 14,681 −64.5 | 20,156 0.44 | 0.03 0 |
| EBIT (millions) | 762 0 | 7492 −85 | 7796 −579 | 6229 −136 | 4066 −102 | 1067 0 | 24,118 −761 | 3326 −1764 | 2628 −516 | 1607 0 |
| Assets (millions) | | 73,913 4.12 | 70,771 3 | 16,570 0 | 655 8.6 | 655 8.6 | 89,993 3 | 37,803 8.6 | 48,779 6.8 | 36,168 0 |
| | | | | | Maxima are listed first in each cell | | | | | |

**Table 1.** *Cont.*

| | | | Panel C | | | | |
|---|---|---|---|---|---|---|---|
| Variable | Excess Return | Market Return | Beta | Market Capitalization | Book to Market | EBIT | Assets |
| Excess Return | 1 | | | | | | |
| Market Return | −0.02 | 1 | | | | | |
| Beta | 0.01 | 0.001 | 1 | | | | |
| Market Capitalization | 0.07 | 0.004 | 0.58 | 1 | | | |
| Book to Market | 0 | $1 \times 10^{-5}$ | 0.33 | 0.002 | 1 | | |
| EBIT | 0.3 | 0.04 | 0.03 | 0.31 | 0.05 | 1 | |
| Assets | 0.002 | 0.15 | 0.04 | 0.30 | 0.04 | 0.93 | 1 |

### 3.2. Methodology

For each year in the 2010–2019 period, daily stock returns, daily S & P 500 (proxy for the market) returns, quarterly beta, and daily Treasury bill rates were collected. Panel data fixed-effects regressions of excess returns as the dependent variable and market returns and beta per year were employed to test the Capital Asset Pricing Model. We support the measurement of returns and predictors over a single year, as we are assuming an ultra-short-term holding period. Investors hold green energy investments for a year, to realize returns at the end of the period. We envision investors as wishing to capitalize on short-term growth opportunities, and then exit the market. Renewable energy investments are novel, and emerging investments. There is a great amount of uncertainty about their long-term performance. Will some of the smaller firms survive? Given this doubt, certain investors may opt for extremely short-term holding periods. It is to the predispositions of these investors that this study is directed. The following equation was used,

$$Excret_t = \beta \left( R_m - R_f \right) + \varepsilon \tag{5}$$

where

*Excret* = daily excess security returns,
$\beta$ = the stock's beta coefficient,
$R_m$ = return on the market portfolio, proxied by the S & P 500, and
$R_f$ = daily Treasury bill rates.

The Fama–French Three-Factor Model was tested by adding size (measured by daily market capitalization of securities) and daily book-to-market values to Equation (5),

$$Excret_t = \beta_1 \left( R_m - R_f \right) + SMB_t(LNME_t) + HML_t(BMKT_t) + \varepsilon \tag{6}$$

$SMB_t$ = size coefficient,
$LNME_t$ = log of market capitalization, and
$HML_t$ = book-to-market coefficient.

The Fama–French Five-Factor Model was tested by adding fixed investment (measured by total assets) and operating income or Earnings before interest and taxes (EBIT) to Equation (6),

$$Excret_t = \begin{array}{l} \beta_1 \left( R_m - R_f \right) + SMB_t(LNME_t) + HML_t(BMKT_t) + \beta_2(INVESTMENT_t) \\ + \beta_3(OPERATING\ INCOME_t) + \varepsilon \end{array} \tag{7}$$

$SMB_t$ = size coefficient,
$LNME_t$ = log of market capitalization,

$HML_t$ = book-to-market coefficient,
$INVESTMENT_t$ = total assets, and
$OPERATING\ INCOME_t$ = Earnings before interest and taxes.

There was a need to prevent endogeneity of independent variables in the Fama–French Three-Factor and Fama–French Five-Factor models. Two-stage least squares can prevent endogeneity. Market capitalization, book-to-market equity, total assets, and operating income were dependent variables in first-stage regressions on predictors including stockholder's equity, size, fixed assets, sales, and gross profit. This was followed by market capitalization, book-to-market equity, total assets and operating income becoming independent variables in their regressions on excess return as the dependent variable, in the second-stage regressions. Lack of significance of predictors in the two stages suggested the absence of the problem of endogeneity of predictors.

The relationship in Equation (6) was subjected to 20th, 50th, and 80th panel quantile regressions, as the 20th, 50th, and 80th quantiles of excess returns are explained by a linear function of market return, beta, size, book-to-market, assets, and EBIT. Quantile regressions are sensitive to outliers, with extreme sensitivity to excess returns in the tails provided by the 20th quantile and the 80th quantile. Equation (7) was adjusted for the inclusion of the two extreme value measures, Value-at-Risk, and Expected Shortfall, as shown in Equations (9) and (10). Quantile regressions at the 20th, 50th, and 80th quantiles of excess returns were performed, with the inclusion of Value-at-Risk and Expected Shortfall, respectively. We justify the use of quantile regression for a sample size of 122, as Haque and Delgado (2017) demonstrated that specification tests in quantile panel regressions observed that sample sizes < 122 (namely, 50 and 100) were undersized and had low power. Power increased as sample sizes were increased above 100. Moreover, it would be challenging to find stocks of renewable energy companies in the largest samples of 800, which yielded the best specification test results.

$$
\begin{aligned}
Excret_t = \ & \beta_1\left(R_m - R_f\right) + SMB_t(LNME_t) + HML_t(BMKT_t) \\
& + \beta_2(INVESTMENT) + \beta_3(OPERATING\ INCOME) + \beta_4\ VAR_t + \varepsilon
\end{aligned}
\tag{8}
$$

$$
\begin{aligned}
Excret_t = \ & \beta_1\left(R_m - R_f\right) + SMB_t(LNME_t) + HML_t(BMKT_t) + \beta_2(INVESTMENT) \\
& + \beta_3(OPERATING\ INCOME) + \beta_4 ES_t + \varepsilon
\end{aligned}
\tag{9}
$$

## 4. Results

### 4.1. Results Pertaining to the Market Model

Hypothesis 1 states that market return and beta, as presented in the single-factor CAPM Model would significantly explain excess returns in green energy securities. As shown in Table 2, Hypothesis 1 was partly supported with significant market returns at $p < 0.001$, in 2010, 2012, 2017, and 2019, and significant beta coefficients in 2011, 2014, and 2017., $p < 0.01$, and $p < 0.001$. Negative betas in 2011 and 2014 suggest that in certain years significantly low systematic risk may explain excess returns. CAPM, with its emphasis on excess returns being explained solely by market returns, as in returns on a market index, such as the S & P 500, does prevail in certain years, showing the domination of large cap green stocks in the portfolio. CAPM's predictability is stronger in the early years, from 2010 to 2012, before the growth in small cap and risky wind, electric vehicle, and solar energy producers. Yet, the explanation of excess returns by market returns and betas is insufficient, as this result does not include size and book-to-market factors that may influence the explanatory power of market returns.. Therefore, we propose and test Hypothesis 2. Hypothesis 2 states that the Fama–French Three-Factor model explained the variation in excess returns. From Table 3, Panel A, Hypothesis 2 was marginally supported, with market returns in 2010 and 2019 significantly explaining excess returns at $p < 0.001$. However, size effects and book-to-market effects eliminated the explanatory power of the beta coefficient, which remained insignificant throughout the period. In other words, size differentials among sampled firms eliminated systematic risk as a predictor of excess

returns. Differences in book to market equity (value versus glamour stocks) also eliminated systematic risk as a predictor of excess returns. It follows that systematic risk or beta may not be a true predictor of excess returns in green energy securities as any systematic risk may be due to differential risk from securities of different size, or differential book to market equity. Hypothesis 3 states that the Fama–French Five-Factor model would explain excess returns. As with other market models, Table 3, Panel B shows that market returns dominated, providing significant explanatory power in 2010, 2012, 2013, and 2019. With the exception of 2010, with a negative beta, EBIT and Assets have eliminated the remaining explanatory power of beta in explaining returns in green energy securities. Was the effect of beta completely eliminated? The response to that query is negative, as at the extreme 20% quantile of excess returns, and 80% quantile of excess returns (see Table 3, Panel C and Panel E), significant reductions in EBIT resulted in significantly negative beta coefficients in selected years. This suggests that when operating income decreases sharply, excess returns are explained by security co-movements in opposite direction to the market. Fama–French predictors of size, book-to-market, investment in fixed assets, and profitability were influential in predicting excess returns, particularly since they nullified the effect of market risk. Green securities are from firms of varying size in terms of market capitalization, and fixed asset investment. Small firms in wind energy production or solar energy production varying visibility as noted by differential book-to-market equity. Less visible firms with superior performance may dominate in market niches, such as certain geographical areas. As this is an international sample, these firms may dominate in certain countries with incentives for reducing the carbon footprint. Startup firms in wind energy production and electric vehicles would have higher fixed investment costs than a large producer who adapts existing production facilities to green energy production. Likewise, green firms varied in operating income and profitability, as certain firms are more efficient in exploiting market opportunities than others. Some of these firms have positive betas, while others have negative betas, which cancel the effect of positive betas on excess returns, thereby rendering beta insignificant in explaining excess returns.

### 4.2. Results Pertaining to the Conditional Extreme Value Distribution

### 4.2.1. Value-at-Risk Results

Hypothesis 4a and Hypothesis 4b stated that in a conditional extreme value distribution, extreme values represented by Value-at-Risk significantly explained excess returns in green energy securities. Hypothesis 4a was fully supported, with the losses that would not be more than the Value-at-Risk levels significantly explaining excess returns in 8 of 10 years. As Table 4, Panel A and Panel B indicate, losses that could not be more than the Value-at-Risk level explained excess returns, regardless of size with 95% confidence. Value-at-Risk explained excess returns in green energy securities for both large and small firm securities. Value-at-Risk also explained excess returns for firms of high or low book-to-market equity with 95% confidence. For extremely low values of excess returns (20% quantile), Value-at-Risk significantly predicted excess returns, as it did for extremely high values of excess returns (80% quantile) with 95% confidence, in 2010, 2016, and 2019 (see Table 4, Panel C and Panel E). At this quantile, Value-at-Risk explained excess returns regardless of levels of fixed investment, or profitability. Hypothesis 4b was fully supported, as indicated by Table 5, Panel A and Panel B. Losses that could not be more than the Value-at-Risk level, explained excess returns, regardless of size, with 99% confidence. Value-at-Risk also explained excess returns for firms of high or low book-to-market equity with 99% confidence. For extremely low values of excess returns (20% quantile), Value-at-Risk significantly predicted excess returns, as it did for extremely high values of excess returns (80% quantile), with 99% confidence, in 2017 and 2019 (see Table 4, Panel C and Panel E). Although Value-at-Risk at both 95% confidence and Value-at-Risk at 99% confidence are significant in explaining excess returns, Value-at-Risk at 95% has greater explanatory power for extreme quantiles of excess return values, in that it is significant for one more year (2017) than Value-at-Risk at 99% confidence. Value-at-Risk represents known losses, or risk that

has certainty. To an investor, risk that is known permits planning. A risk-averse investor would avoid investments in green securities that generate losses beyond the Value-at-Risk level. A risk-taker would be willing to take a chance on investments that are riskier than the Value-at-Risk level. Value-at-Risk is the demarcation between risky investments with certainty of risk and highly risky investments with uncertainty of risk.

4.2.2. Expected Shortfall Results

Hypothesis 5 posited that Expected Shortfall would significantly explain the variation in excess returns in green energy securities. Hypothesis 6 was fully supported for the Fama–French Three-Factor Model, and 20th quantile of excess returns (see Table 6, Panel A and Panel C). Expected Shortfall significantly reduced excess returns in 2012, 2015, and 2019, with the inclusion of size and book-to-market differentials, suggesting that losses beyond the Value-at-Risk percentile could reduce excess returns in green energy securities, for both large and small firms, and value and glamour stocks. Thus, Expected Shortfall complements the explanatory power of Value-at-Risk. Value-at-risk indicates that losses at the 95th or 99th percentile reduce excess returns, while Expected Shortfall indicates that average losses beyond this level reduce excess returns. For extremely low 20th quantile excess returns, Expected Shortfall significantly reduced returns in 2015 and 2019, indicating that extreme values may have more powerful effects on extremely low returns. In Section 4.2.1, we presented Value-at-Risk as the demarcation between certain risk and uncertain risk. Risk-takers may wish to invest in green energy producers with uncertain risk. They could use our Expected Shortfall values to assess the level of losses beyond the Value-at-Risk level.

Examination of skewness and kurtosis values showed a positively skewed distribution with significant coefficients of skewness for each year in the sample period. The significance of kurtosis values indicated flatness, with heavy tails. Jarque–Bera tests indicated non-normality for each year of the study period. Significantly positive kurtosis values for excess returns for each year of the sample period provide evidence of a flat leptokurtic distribution. Given the significance of Expected Shortfall in certain years that averages the losses during those years in percentiles beyond the Value-at-Risk percentiles, we may assume the presence of excess returns in the tails, or the presence of heavy tails.

**Table 2.** Predictors of daily excess returns using CAPM.

| Independent Variables (CAPM Predictors of Excess Return) | 2010 | 2011 | 2012 | 2013 | 2014 | 2015 | 2016 | 2017 | 2018 | 2019 |
|---|---|---|---|---|---|---|---|---|---|---|
| Market Return | 3.57 *** | 5.15 | 3.32 * | 0.93 | −4.93 | −0.91 | −1.05 | 3.86 *** | 2.74 | −1.41 * |
| Beta | $2.28 \times 10^{-3}$ | −0.45 *** | $7 \times 10^3$ | 0.01 * | −2.5 ** | $-3.38 \times 10^{-2}$ | $9 \times 10^{-3}$ | 0.06 *** | −0.05 | $1.51 \times 10^{-2}$ |
| N | 16,485 | 12,269 | 15,539 | 12,756 | 13458 | 16,993 | 13,482 | 13,536 | 13,712 | 14,073 |
| $R^2$ | $1.6 \times 10^{-2}$ | 0.02 | $1.6 \times 10^{-4}$ | 0.04 | 0.019 | $1.56 \times 10^{-2}$ | 0.03 | 0.02 | 0.01 | $1.75 \times 10^2$ |

\* $p < 0.05$, \*\* $p < 0.01$, \*\*\* $p < 0.001$.

**Table 3.** Panel A. Predictors of daily excess returns using the Three-Factor Fama–French Model, 20th, 50th, and 80th quantiles. Panel B. Predictors of daily excess returns the Five-Factor Fama–French Model fixed-effects panel data regressions. Dependent variable = excess return. Cell values are regression coefficients. Panel C. Predictors of daily excess returns using quantile regression with 20% quantiles. Panel D. Predictors of daily excess returns using quantile regression with 50% quantiles. Panel E. Predictors of daily excess returns using quantile regression with 80% quantiles.

| Panel A | | | | | | | | | | |
|---|---|---|---|---|---|---|---|---|---|---|
| **Independent Variable** | **2010** | **2011** | **2012** | **2013** | **2014** | **2015** | **2016** | **2017** | **2018** | **2019** |
| CAPM Predictor | | | | | | | | | | |
| Market Return | 3.06 *** | 6.62 | 4.13 * | 1.54 ** | −8.31 | −3.32 | −0.94 | 3.18 | 3.48 | −2.67 ** |
| Beta | $3.26 \times 10^{-3}$ * | $6 \times 10^{-2}$ | 0.09 | $3.88 \times 10^{-3}$ | $-2.36 \times 10^{-3}$ | $-4 \times 10^{-2}$ | 0.03 | 0.05 | 0.01 | $8.8 \times 10^{-3}$ |
| Fama–French Predictor | | | | | | | | | | |
| Market Capitalization | $-7.8 \times 10^{-3}$ | $-8.8 \times 10^{-4}$ | $7.31 \times 10^{-3}$ | $2.46 \times 10^{-3}$ | $5.51 \times 10^{-3}$ | $3.01 \times 10^{-2}$ | $5.67 \times 10^{-3}$ | $1.96 \times 10^{-5}$ | $9.34 \times 10^3$ | $4.23 \times 10^{-3}$ |
| Book-to-Market | −8.44 | $6.3 \times 10^{-5}$ | $1.16 \times 10^{-5}$ | 0.00 | $-1.69 \times 10^{-5}$ | 2.11 | $10.41 \times 10^{-5}$ | $4.61 \times 10^{-5}$ | $11.25 \times 10^{-6}$ | 5.74 |
| N | 16,933 | 9010 | 8372 | 7943 | 7601 | 17,811 | 7353 | 6299 | 6987 | 14,316 |
| $R^2$ | 0.05 | 0.04 | 0.03 | 0.08 | 0.03 | $1.52 \times 10^{-2}$ | 0.05 | 0.17 | 0.02 | $1.77 \times 10^{-2}$ |

**Table 3.** *Cont.*

| | | | | | Panel B | | | | | |
|---|---|---|---|---|---|---|---|---|---|---|
| **Independent Variable** | **2010** | **2011** | **2012** | **2013** | **2014** | **2015** | **2016** | **2017** | **2018** | **2019** |
| CAPM Predictor | | | | | | | | | | |
| Market Return | 1.03 *** | 6.71 | 4.14 * | 1.54 ** | −8.32 | −10.31 | −0.93 | 3.17 | 3.47 | −2.67$^8$ |
| Beta | −0.03 *** | 0.05 | 0.09 | $4.34 \times 10^{-3}$ | $4.08 \times 10^{2}$ | −0.82 | 0.03 | 0.03 | $6.18 \times 10^{-3}$ | $9.03 \times 10^{-3}$ |
| Fama–French Predictor | | | | | | | | | | |
| Market Capitalization | $5.3 \times 10^{-3}$ *** | $-2.9 \times 10^{-3}$ | $8.04 \times 10^{-3}$ | $2.59 \times 10^{-3}$ | $-1.35 \times 10^{-3}$ | −0.06 | $3.62 \times 10^{-3}$ | $1.38 \times 10^{-3}$ | $1.32 \times 10^{-2}$ | $4.18 \times 10^{-3}$ |
| Book-to-Market | −88.38 *** | $-1.05 \times 10^{-5}$ | $1.99 \times 10^{-5}$ | $1.02 \times 10^{-5}$ | $1.97 \times 10^{-3}$ | −62.79 | $7.43 \times 10^{-4}$ | $1.3 \times 10^{-4}$ | $2.22 \times 10^{-4}$ | 5.75 |
| EBIT | $2.38 \times 10^{6}$ | $1.34 \times 10^{-4}$ | $-1.50 \times 10^{-5}$ | $7.61 \times 10^{-6}$ | $-6.67 \times 10^{-5}$ | $-3.97 \times 10^{-4}$ | $3.96 \times 10^{-6}$ | $-5.35 \times 10^{-6}$ | $-3.54 \times 10^{-5}$ | |
| Assets | $2.90 \times 10^{-6}$ | $1.16 \times 10^{-6}$ | $7.03 \times 10^{-7}$ | $-4.90 \times 10^{-7}$ | $-8.14 \times 10^{-5}$ | $-2.39 \times 10^{-5}$ | $-3.03 \times 10^{-5}$ | $-5.33 \times 10^{-6}$ | $-9.12 \times 10^{-6}$ | $2.27 \times 10^{-6}$ |
| N | 3,534 | 8948 | 8372 | 7943 | 7601 | 5,744 | 7353 | 6299 | 6987 | 14,316 |
| $R^2$ | 0.18 | 0.04 | 0.03 | 0.08 | 0.03 | 0.05 | 0.05 | 0.17 | 0.02 | $1.77 \times 10^{2}$ |
| | | | | | Panel C | | | | | |
| **Independent Variable** | **2010** | **2011** | **2012** | **2013** | **2014** | **2015** | **2016** | **2017** | **2018** | **2019** |
| CAPM Predictor | | | | | | | | | | |
| Market Return | 1.13 | 43.75 *** | 40.9 *** | −102 | 9.20 | −0.51 | 0.52 | −.12 | −1.13 | 4.39 |
| Beta | 0.09 | −5.91 *** | −4.49 *** | −0.06 | $7.2 \times 10^{-2}$ * | $-2.79 \times 10^{-3}$ | $-7.5 \times 10^{-3}$ | −.06** | −.03 | −12.51 |
| Fama–French Predictor | | | | | | | | | | |
| Market Capitalization | 0.05 | $-3.9 \times 10^{-3}$ | $-1.4 \times 10^{-2}$ * | $5.77 \times 10^{-3}$ | $7.65 \times 10^{-3}$ * | $-1.23 \times 10^{-3}$ | $8.55 \times 10^{-5}$ | $2.60 \times 10^{-3}$ * | $3.11 \times 10^{-3}$ | 0.33 |
| Book-to-Market | −63.9 | $-9.96 \times 10^{-6}$ | $1 \times 10^{-4}$ | $2.60 \times 10^{-4}$ | $6.89 \times 10^{3}$ ** | 0.10 | $6.53 \times 10^{-4}$ | $2.12 \times 10^{-4}$ | $4.25 \times 10^{-4}$ | 46.85 |
| EBIT | $9.01 \times 10^{-4}$ | $5.60 \times 10^{-4}$ | $1.96 \times 10^{-3}$ *** | $5.79 \times 10^{-4}$ | $1.17 \times 10^{3}$ ** | $-2 \times 10^{-6}$ | $-5.24 \times 10^{-4}$ *** | $-6.82 \times 10^{-4}$ *** | $-9.58 \times 10^{-4}$ *** | |
| Assets | $1.4 \times 10^{-2}$ | $1.14 \times 10^{-5}$ | $5.28 \times 10^{-5}$ *** | $-1.0 \times 10^{-5}$ | $-3.1 \times 10^{-4}$ *** | $4.8 \times 10^{-6}$ | $-2.78 \times 10^{-5}$ | $1.43 \times 10^{-5}$ * | $-3.3 \times 10^{-6}$ | −0.67 *** |
| N | 3534 | 8948 | 8372 | 7943 | 7601 | 5744 | 7353 | 6299 | 6987 | 14,316 |

**Table 3.** *Cont.*

| Panel D | | | | | | | | | | |
|---|---|---|---|---|---|---|---|---|---|---|
| **Independent Variable** | **2010** | **2011** | **2012** | **2013** | **2014** | **2015** | **2016** | **2017** | **2018** | **2019** |
| CAPM Predictor | | | | | | | | | | |
| Market Return | 3.66 | 45.10 *** | −31 *** | 102 | 20 *** | 0.50 | 0.53 | 0.80 | −2.22 | 5.47 |
| Beta | −1.73*** | −5.24 *** | 0.1 *** | 0.24 | −0.02 | −8.4 × $10^{-4}$ | 8.55 × $10^{-3}$ | −0.04 ** | −0.05 | −9.36 |
| Fama–French Predictor | | | | | | | | | | |
| Market Capitalization | −0.07 | 2.28 × $10^{-2}$ *** | −9.4 × $10^{-3}$ * | 0.07 | 0.06 *** | 2.42 × $10^{-5}$ | −4.89 × $10^{-4}$ | 8.34 × $10^{-4}$ | 1.07 × $10^{-3}$ | −0.70 |
| Book-to-Market | 20.03 | −3.40 × $10^{-4}$ | 7.4 × $10^{-4}$ *** | 1.3 × $10^{-5}$ | −5.2 × $10^{-4}$ | 0.03 | −2.12 × $10^{-6}$ | −3.55 × $10^{-5}$ | −7.29 × $10^{-5}$ | 12.40 |
| EBIT | −1.2 × $10^{-3}$ | 2.4 × $10^{-4}$ | −1.4 × $10^{3}$ *** | 1.35 × $10^{-3}$ | 6.68 × $10^{-4}$ ** | 4 × $10^{-6}$ | −7.87 × $10^{-5}$ *** | −1.1 × $10^{-3}$ *** | 7.11 × $10^{-5}$ | |
| Assets | −0.02*** | 1.31 × $10^{-5}$ | 5.6 × $10^{-7}$ | −9 × $10^{-6}$ | −3.3 × $10^{-5}$ | 3.0 × $10^{-6}$ | 1.8 × $10^{-7}$ | 7.1 × $10^{-7}$ | 8.44 × $10^{-6}$ | 9.31 *** |
| N | 3534 | 8948 | 8372 | 7943 | 7601 | 5.44 | 7353 | 6299 | 6987 | 14,316 |
| Panel E | | | | | | | | | | |
| **Independent Variable** | **2010** | **2011** | **2012** | **2013** | **2014** | **2015** | **2016** | **2017** | **2018** | **2019** |
| CAPM Predictor | | | | | | | | | | |
| Market Return | 2.66 | 49.14 *** | −51 *** | 117 | 22 [8]** | 0.45 | 0.56 | 0.23 | −2.00 | 2.56 |
| Beta | 2.90 *** | −1.92 *** | 0.9 *** | 0.28 | 0.1 ** | 2.30 × $10^{-3}$ | 0.01 | −7.35 × $10^{-3}$ | −0.03 | −5.50 |
| Fama–French Predictor | | | | | | | | | | |
| Market Capitalization | −0.02 | 9.66 *** | 2.1 × $10^{-3}$ | 0.18 | 0.10 * | 1.11 × $10^{-3}$ | 1.37 × $10^{-3}$ | −2.34 × $10^{-3}$ | −2.06 × $10^{-3}$ | −1.38 |
| Book-to-Market | 48.37 | −5.33 × $10^{-4}$ ** | 4.53 × $10^{-4}$ *** | 3.22 × $10^{-4}$ | 4.8 × $10^{5}$ | 0.43 | −1.42 × $10^{-4}$ | −4.42 × $10^{-5}$ | −4.69 × $10^{-4}$ | 49.13 |
| EBIT | −8.1 × $10^{-4}$ | −1.21 × $10^{-3}$ ** | 1.4 × $10^{3}$ *** | −1 × $10^{3}$ | −8.16 × $10^{-4}$ *** | −6.7 × $10^{-6}$ | −6.48 × $10^{-5}$ *** | −4.61 × $10^{-6}$ | −6.06 × $10^{-4}$ *** | |
| Assets | −8.04 × $10^{-3}$ | 3.8 × $10^{-5}$ * | −4.9 × $10^{-5}$ *** | 2.6 × $10^{-5}$ | −1.61 × $10^{-5}$ | 8.2 × $10^{-6}$ | 7.08 × $10^{-6}$ | 9.42 × $10^{-6}$ | 3.48 × $10^{-5}$ * | −6.29 *** |
| N | 3534 | 8948 | 8372 | 7943 | 7601 | 5,744 | 7353 | 6299 | 6987 | 14,316 |

* $p < 0.05$, ** $p < 0.01$, *** $p < 0.001$.

**Table 4.** Panel A. Predictors of daily excess returns with Value-at-Risk 95 and Three Fama–French Factors, Five Fama-French Factors, 20[th], 50[th], and 80[th] quantile Value-at-Risk 95 factors. Panel B. Predictors of daily excess returns with Value-at-Risk 95 and Five Fama–French Factors. Panel C. Predictors of daily excess returns with Value-at-Risk with 20% quantiles. Panel D. Predictors of daily excess returns with Value-at-Risk using 50% quantiles. Panel E. Predictors of daily excess returns with Value-at-Risk 95 with 80% quantiles.

| | | | | | Panel A | | | | | |
|---|---|---|---|---|---|---|---|---|---|---|
| **Independent Variable** | **2010** | **2011** | **2012** | **2013** | **2014** | **2015** | **2016** | **2017** | **2018** | **2019** |
| CAPM Predictor | | | | | | | | | | |
| Market Return | 3.3 *** | 7.09 | 3.63 | 1.7 *** | −1.29 | −2.41 | −1.27 | 3.18 | 3.28 | −3.08 |
| Beta | $3.1 \times 10^{-3}$ * | $6.65 \times 10^{-2}$ | 0.05 | 0.02 | $-1.08 \times 10^{-3}$ | −0.06 | −0.06 | 0.04 | −0.01 | 0.00 |
| Fama–French Predictor | | | | | | | | | | |
| Market Capitalization | $-4.7 \times 10^{3}$ | $5.06 \times 10^{-3}$ | $4.7 \times 10^{-3}$ | $2.9 \times 10^{-3}$ | $7.1 \times 10^{-3}$ | $-8.24 \times 10^{-3}$ | 0.02 ** | $9.4 \times 10^{-5}$ | $8.1 \times 10^{-3}$ | $-6 \times 10^{-3}$ |
| Book-to-Market | −18.9 | $1.06 \times 10^{-4}$ | $4.4 \times 10^{-5}$ | 0.00 | $-2.3 \times 10^{-5}$ | −3.23 | $1.5 \times 10^{-5}$ | $4.8 \times 10^{-6}$ | $-1.0 \times 10^{-6}$ | 10.65 |
| Extreme Value Distribution Predictor | | | | | | | | | | |
| Value-at-Risk 95 | $-1.3 \times 10^{-5}$ *** | $5.54 \times 10^{-6}$ *** | $-6.7 \times 10^{-6}$ *** | −0.00 | $-6.4 \times 10^{-5}$ *** | $-6.73 \times 10^{-6}$ *** | $-6.7 \times 10^{-6}$ *** | $-5.9 \times 10^{-7}$ | $-3.4 \times 10^{-6}$ *** | $-6.67 \times 10^{-6}$ *** |
| N | 16,470 | 8285 | 8056 | 6897 | 7326 | 17,944 | 7010 | 6299 | 7025 | 13,659 |
| R$^2$ | 0.09 | 0.05 | 0.07 | 0.12 | 0.06 | 0.04 | 0.08 | 0.17 | 0.04 | 0.04 |
| | | | | | Panel B | | | | | |
| **Independent Variable** | **2010** | **2011** | **2012** | **2013** | **2014** | **2015** | **2016** | **2017** | **2018** | **2019** |
| CAPM Predictor | | | | | | | | | | |
| Market Return | 1.3 *** | 7.21 | 3.64 | 1.7 *** | −6.05 | −4.42 | −1.25 | 3.18 | 3.2 | −3.08 |
| Beta | −0.07 *** | 0.03 | 0.05 | 0.02 | $5.1 \times 10^{-3}$ | 0.02 | −0.05 | 0.03 | −0.01 | 0.00 |
| Fama–French Predictor | | | | | | | | | | |
| Market Capitalization | $3.4 \times 10^{-3}$ | $3.1 \times 10^{-4}$ | $7.2 \times 10^{-3}$ | $4.4 \times 10^{-3}$ * | $6.5 \times 10^{-3}$ | $-2.45 \times 10^{-3}$ | 0.02 ** | $1.3 \times 10^{-3}$ | $9.2 \times 10^{-3}$ | $-6 \times 10^{-3}$ |
| Book-to-Market | −26.8 | $-1.3 \times 10^{-6}$ | $6.8 \times 10^{-5}$ | $2.5 \times 10^{-5}$ | $8.4 \times 10^{-4}$ | −24.50 | $1.2 \times 10^{-3}$ | $1.2 \times 10^{-4}$ | $8.5 \times 10^{-6}$ | 10.67 |
| EBIT | $-3.1 \times 10^{-5}$ | $2.3 \times 10^{-4}$ | $-8.6 \times 10^{-6}$ | $1.6 \times 10^{-5}$ | $-1.5 \times 10^{-5}$ | $-2.45 \times 10^{-5}$ | $1.6 \times 10^{-6}$ | $-4.5 \times 10^{-6}$ | $3.0 \times 10^{-5}$ | $-6.34 \times 10^{-6}$ |
| Assets | | $1.2 \times 10^{-6}$ | $-2.4 \times 10^{-6}$ | $-1.2 \times 10^{6}$ | $-3.5 \times 10^{-5}$ | | | $-4.8 \times 10^{-5}$ | $-4.9 \times 10^{-6}$ | $-3.6 \times 10^{-6}$ | $3.75 \times 10^{-6}$ |
| Extreme Value Distribution Predictor | | | | | | | | | | |
| Value-at-Risk 95 | $-3.3 \times 10^{-6}$ | $5.5 \times 10^{-6}$ *** | $-6.7 \times 10^{-6}$ *** | $-4.1 \times 10^{-8}$ | $-6.4 \times 10^{-6}$ *** | $-7 \times 10^{-6}$ *** | $-6.7 \times 10^{-6}$ *** | $-5.8 \times 10^{-7}$ | $-3.4 \times 10^{-6}$ *** | $-6.67 \times 10^{-6}$ *** |
| N | 4767 | 8223 | 8056 | 6897 | 7326 | 12334 | 7010 | 6299 | 7025 | 13,659 |
| R$^2$ | 0.05 | 0.05 | 0.07 | 0.12 | 0.06 | 0.04 | 0.08 | 0.17 | 0.04 | 0.04 |

**Table 4.** *Cont.*

| | | | | | Panel C | | | | | |
|---|---|---|---|---|---|---|---|---|---|---|
| **Independent Variable** | **2010** | **2011** | **2012** | **2013** | **2014** | **2015** | **2016** | **2017** | **2018** | **2019** |
| CAPM Predictor | | | | | | | | | | |
| Market Return | −0.03 | 45.5 *** | −39.4 *** | −100 | 108 ** | 0.49 | 36.4 | 0.46 | −1.05 | 8.4 |
| Beta | −0.01 | 0.24 ** | −4.8 *** | −1.38 | 2.6 *** | −0.01 | −26 *** | −0.07 *** | $−1.3 \times 10^{−3}$ | 4.10 |
| Fama–French Predictor | | | | | | | | | | |
| Market Capitalization | $6.48 \times 10^{−3}$ | −0.08 *** | 0.03 *** | −0.15 | −0.49 *** | $−6.07 \times 10^{−5}$ | −0.1 *** | $2.5 \times 10^{−3}$ * | $1.7 \times 10^{−3}$ | 1.08 |
| Book-to- Market | 51.2 | $−4.2 \times 10^{−5}$ | $−3 \times 10^{−4}$ * | $7.04 \times 10^{−4}$ | 0.03 * | −0.90 | 0.09 *** | $1 \times 10^{−4}$ | $1.2 \times 10^{−3}$ ** | 15.96 |
| EBIT | $6.7 \times 10^{−4}$ *** | $−2.4 \times 10^{−4}$ | $3 \times 10^{−3}$ *** | $−8 \times 10^{−6}$ | $6.5 \times 10^{−3}$ *** | $−6.4 \times 10^{−6}$ | $1.2 \times 10^{−3}$ *** | $−8 \times 10^{4}$ *** | $−2\ 10^{−4}$ *** | 19.17 *** |
| Assets | | $1.4 \times 10^{−5}$ | $4.1 \times 10^{−5}$ *** | $7.3 \times 10^{−5}$ | $−1.2 \times 10^{−3}$ ** | | $−3.7 \times 10^{−3}$*** | $−1.9 \times 10^{−5}$ ** | $−3.6 \times 10^{−5}$ * | −13.75 *** |
| Extreme Value Distribution | | | | | | | | | | |
| Value-at-Risk 95 | $6 \times 10^{−5}$ * | $−6 \times 10^{−8}$ | $1.6 \times 10^{−7}$ | $1 \times 10^{−6}$ | $−4 \times 10^{−7}$ | −0.00 | $−4.1 \times 10^{−5}$ ** | $1.8 \times 10^{−6}$ ** | $−1.7 \times 10^{−8}$ | $5.56 \times 10^{−3}$ *** |
| N | 4767 | 8223 | 8056 | 6897 | 7326 | 12,334 | 7010 | 6299 | 7025 | 13,572 |
| | | | | | Panel D | | | | | |
| **Independent Variable** | **2010** | **2011** | **2012** | **2013** | **2014** | **2015** | **2016** | **2017** | **2018** | **2019** |
| CAPM Predictor | | | | | | | | | | |
| Market Return | −0.03 | 52.2 *** | −31 *** | −59.6 | 109 *** | 0.46 | 18.5 | −0.2 | −1.8 | 8.4 |
| Beta | 1.29 *** | 1.5 *** | −1.4 ** | −6.40 | 10 *** | $−9.96 \times 10^{−6}$ | −17 *** | −0.03 ** | $−3 \times 10^{−3}$ | 4.1 |
| Fama–French Predictor | | | | | | | | | | |
| Market Capitalization | $−1 \times 10^{−3}$ | −0.03 *** | 0.02 *** | $−8.3 \times 10^{−3}$ | −0.31 *** | $−9.1 \times 10^{−6}$ | −0.26 *** | $7.9 \times 10^{−4}$ | $−1 \times 10^{−3}$ | 1.08 |
| Book-to-Market | −11.74 | $−3.6 \times 10^{−5}$ | $−1.3 \times 10^{−4}$ | $−2.7 \times 10^{−5}$ | $−1.2 \times 10^{−3}$ | 0.08 | $6 \times 10^{−3}$ | $−1.5 \times 10^{−4}$ | $2.7 \times 10^{−4}$ | 15.96 |
| EBIT | $6.4 \times 10^{−4}$*** | $1.1 \times 10^{−4}$ | $−4.6 \times 10^{−4}$ | $2.3 \times 10^{−3}$ | $4.9 \times 10^{−3}$ *** | $7.3 \times 10^{−7}$ | $2.2 \times 10^{−4}$ * | $−7.4 \times 10^{−4}$ *** | $7.9 \times 10^{−4}$ *** | 19.17 *** |
| Assets | | $9.2 \times 10^{−6}$ | $−3.7 \times 10^{−5}$ *** | $2.6 \times 10^{−5}$ | $−1.2 \times 10^{−4}$ | | $−4 \times 10^{−4}$ | $8.4 \times 10^{−6}$ | $−7 \times 10^{−6}$ | −13.75 *** |
| Extreme Value Distribution Predictor | | | | | | | | | | |
| Value-at-Risk 95 | $8.66 \times 10^{−6}$ | $3.2 \times 10^{−7}$ | $3.2 \times 10^{−7}$ | −0.00 | $−3.2 \times 10^{−6}$ | 0.00 | $−3.4 \times 10^{−5}$ ** | $3.6 \times\times 10^{−8}$ | $7.9 \times 10^{−8}$ | $5.56 \times 10^{−3}$ *** |
| N | 4767 | 8223 | 8056 | 6897 | 7326 | 12,334 | 7010 | 6299 | 7025 | 13,572 |

**Table 4.** *Cont.*

| | | | | | Panel E | | | | | |
|---|---|---|---|---|---|---|---|---|---|---|
| **Independent Variable** | **2010** | **2011** | **2012** | **2013** | **2014** | **2015** | **2016** | **2017** | **2018** | **2019** |
| CAPM Predictor | | | | | | | | | | |
| Market Return | 0.09 | 52.5 *** | −43 *** | $-5 \times 10^{-3}$ | 104 ** | 0.44 | 20.9 | 0.18 | −1.8 | 8.11 |
| Beta | 1.27 *** | 2.8 *** | 1.0 ** | −2.8 | 18 *** | $8.59 \times 10^{-2}$ | −5.2 *** | −0.02 * | 0.05 * | 6.68 |
| Fama–French Predictor | | | | | | | | | | |
| Market Capitalization | $-1 \times 10^{-3}$ | −0.01 | 0.05 *** | −0.02 | −0.04 * | $1.63 \times 10^{-4}$ | −0.1*** | $-3.2 \times 10^{3}$ ** | $-2 \times 10^{-3}$ | −0.14 |
| Book-to-Market | −43.7 | $-3.8 \times 10^{-4}$ * | $8.6 \times 10^{-4}$*** | $-4.4 \times 10^{-4}$ | −0.04 *** | .99 | −0.03 ** | $-3.1 \times 10^{-4}$ * | $-1.5 \times 10^{-4}$ | 21.14 |
| EBIT | $-6.3 \times 10^{-5}$ | $-9.4 \times 10^{-4}$ | $2.3 \times 10^{-3}$ *** | $2.9 \times 10^{-3}$ | $6.1 \times 10^{-3}$ | $5.73 \times 10^{-6}$ | $1 \times 10^{-3}$ *** | $-6.9 \times 10^{-5}$ *** | $5.2 \times 10^{-4}$ *** | 10.50 *** |
| Assets | | $3.7 \times 10^{-5}$ * | $-2.8 \times 10^{-5}$ * | $4.2 \times 10^{-5}$ | $1.7 \times 10^{-3}$ *** | | $1.6 \times 10^{-3}$ ** | $2.0 \times 10^{-5}$ *** | $2.2 \times 10^{-5}$ | −6.70 *** |
| Extreme Value Distribution Factor | | | | | | | | | | |
| Value-at-Risk 95 | $-5.2 \times 10^{-5}$ * | $-4.9 \times 10^{-7}$ | $-7 \times 10^{-7}$ | $1 \times 10^{-6}$ | $-2.6 \times 10^{-6}$ | 0.00 | $-1.4 \times 10^{-5}$ | $-6.9 \times 10^{-7}$ | $-3.4 \times 10^{-7}$ | $1 \times 10^{-2}$ *** |
| N | 4767 | 8223 | 8056 | 6897 | 7326 | 12334 | 7010 | 6299 | 7025 | 13,572 |

$* p < 0.05, ** p < 0.01, *** p < 0.001.$

**Table 5.** Panel A. Predictors of daily excess returns with Value-at-Risk 99 and Three Fama–French Factors, Five Fama-French factors, 20th, 50th, and 80th quantiles. Panel B. Predictors of daily excess returns using with Value-at-Risk 99 with Five Fama–French Factors. Panel C. Predictors of daily excess returns with Value-at-Risk 99 and 20% quantiles. Panel D. Predictors of daily excess returns with Value-at-Risk 99 with 50% quantiles. Panel E. Predictors of daily excess returns with Value-at-Risk 99 and 80% quantiles.

| | | | | | Panel A | | | | | |
|---|---|---|---|---|---|---|---|---|---|---|
| **Independent Variable** | **2010** | **2011** | **2012** | **2013** | **2014** | **2015** | **2016** | **2017** | **2018** | **2019** |
| CAPM Predictor | | | | | | | | | | |
| Market Return | 3.3 *** | 6.67 | 3.48 | 1.55 ** | −6.0 | −2.41 | −0.64 | 3.18 | 3.25 | −2.9 *** |
| Beta | $3.1 \times 10^{-3}$ * | $-1.9 \times 10^{-3}$ | 0.02 | $3.9 \times 10^{3}$ | $3.1 \times 10^{-3}$ | −0.06 | 0.01 | 0.04 | $-2.2 \times 10^{-3}$ | $7.3 \times 10^{-3}$ |
| Fama–French Factor | | | | | | | | | | |
| Market Capitalization | $-4.7 \times 10^{-3}$ | $-2.7 \times 10^{-3}$ | $5 \times 10^{-3}$ | $2.4 \times 10^{-3}$ | $3.5 \times 10^{-3}$ | $-8 \times 10^{-3}$ | $6.1 \times 10^{-3}$ | $9.9 \times 10^{-5}$ | $5.3 \times 10^{-3}$ | $-5 \times 10^{-3}$ |
| Book-to-Market | −18.7 | $3 \times 10^{-6}$ | $4.6 \times 10^{-5}$ | 0.00 | $-2 \times 10^{-5}$ | −3.23 | $-6.6 \times 10^{-6}$ | $4.8 \times 10^{-6}$ | $-1.2 \times 10^{-6}$ | 9.8 |
| Extreme Value Distribution Factor | | | | | | | | | | |
| Value-at-Risk 99 | $-1 \times 10^{-5}$ *** | $-7.1 \times 10^{-6}$ *** | $-5.6 \times 10^{-7}$ *** | 0.00 | $-6.4 \times 10^{-6}$ *** | $-5.6 \times 10^{-6}$ *** | $-5.5 \times 10^{-6}$ *** | $-5 \times 10^{-8}$ | $-2.9 \times 10^{6}$ *** | $-5.6 \times 10^{-6}$ *** |
| N | 16,874 | 9013 | 8372 | 8007 | 7601 | 17,945 | 7354 | 6299 | 7025 | 14,319 |
| $R^2$ | 0.09 | 0.07 | 0.07 | 0.08 | 0.06 | 0.04 | 0.07 | 0.17 | 0.04 | 0.04 |
| | | | | | Panel B | | | | | |
| **Independent Variable** | **2010** | **2011** | **2012** | **2013** | **2014** | **2015** | **2016** | **2017** | **2018** | **2019** |
| Market Return | 1.3 *** | 6.75 | 3.49 | $1.54^{8}$* | −6.04 | −4.42 | −0.64 | 3.18 | 3.2 | −2.9 *** |
| CAPM Predictor | | | | | | | | | | |
| Beta | −0.07 *** | $-2.7 \times 10^{-3}$ | 0.02 | $4.3 \times 10^{-3}$ | 0.02 | −0.02 | 0.01 | 0.03 | $1.1 \times 10^{-3}$ | $7.4 \times 10^{-3}$ |
| Market Capitalization | $3 \times 10^{-3}$ | $-3.4 \times 10^{-3}$ | $7.1 \times 10^{-3}$ | $2.4 \times 10^{-3}$ | $1.6 \times 10^{-3}$ | $-2 \times 10^{-3}$ | $4.4 \times 10^{-3}$ | $1.3 \times 10^{-3}$ | $5.9 \times 10^{-3}$ | $-5.1 \times 10^{-3}$ |
| Book-to-Market | −26.2 | $-1.9 \times 10^{-5}$ | $6.8 \times 10^{-5}$ | $1.0 \times 10^{-5}$ | $5.1 \times 10^{-4}$ | −24.5 | $6.1 \times 10^{-4}$ | $1.2 \times 10^{-4}$ | $5.0 \times 10^{-5}$ | 9.1 |
| EBIT | $-3 \times 10^{-5}$ | $1.7 \times 10^{-4}$ | $-6.8 \times 10^{-6}$ | $7.5 \times 10^{-6}$ | $-1.8 \times 10^{-5}$ | $-2.4 \times 10^{-4}$ | $4.2 \times 10^{-5}$ | $-4.5 \times 10^{-6}$ | $2.3 \times 10^{-5}$ | $6.2 \times 10^{-6}$ |
| Assets | | $7.6 \times 10^{-7}$ | $-2.3 \times 10^{-5}$ | $-4.8 \times 10^{-7}$ | $-2.1 \times 10^{-5}$ | | $-2.5 \times 10^{-5}$ | $-4.9 \times 10^{-6}$ | $-2.1 \times 10^{-6}$ | $4.5 \times 10^{-7}$ |
| Extreme Vslue Distribution Predictor | | | | | | | | | | |
| Value-at-Risk 99 | $-2.7 \times 10^{-6}$ | $-7.1 \times 10^{-6}$ *** | $-5.6 \times 10^{-6}$ *** | $1 \times 10^{-9}$ | $-6.4 \times 10^{-6}$ *** | $-6 \times 10^{-6}$ *** | $-5.5 \times 10^{-6}$ *** | $-4.8 \times 10^{-7}$ | $-2.9 \times 10^{-6}$ *** | $-5.6 \times 10^{-6}$ *** |
| N | 4850 | 8951 | 8372 | 8007 | 7601 | 12335 | 7354 | 6299 | 7025 | 14,319 |
| $R^2$ | 0.05 | 0.07 | 0.07 | 0.08 | 0.06 | 0.08 | 0.07 | 0.17 | 0.04 | 0.04 |

**Table 5.** *Cont.*

|  | Panel C | | | | | | | | | |
|---|---|---|---|---|---|---|---|---|---|---|
| **Independent Variable** | **2010** | **2011** | **2012** | **2013** | **2014** | **2015** | **2016** | **2017** | **2018** | **2019** |
| CAPM Predictor | | | | | | | | | | |
| Market Return | −0.12 | 42 *** | −42.5 *** | −23.4 | 18 *** | 0.49 | 0.92 | −0.12 | −0.5 | 8.08 |
| Beta | −0.20 | −0.4 *** | −2.3 *** | −1.2 | −0.05 | −0.01 | 0.01 | −0.06 *** | −0.05 | 2.15 |
| Fama–French Predictor | | | | | | | | | | |
| Market Capitalization | $6.7 \times 10^{-4}$ | −.06*** | −0.04 *** | .21 | $6.6 \times 10^{-3}$ * | $-6 \times 10^{-5}$ | $2.5 \times 10^{-3}$ * | $4.8 \times 10^{-3}$ *** | $9.4 \times 10^{-4}$ | 1.29 |
| Book-to-Market | 44.2 | $2.5 \times 10^{-4}$ | $2.3 \times 10^{-4}$ | $-4.1 \times 10^{-4}$ | $8.2 \times 10^{3}$ *** | −0.9 | $3.3 \times 10^{-4}$ | $4.2 \times 10^{-5}$ | $6 \times 10^{-4}$ | 54.9 |
| EBIT | $3.7 \times 10^{-4}$ ** | $-2.8 \times 10^{-4}$ | $1.1 \times 10^{-3}$ ** | $5.8 \times 10^{-3}$ | $-1.0 \times 10^{-3}$ *** | $-6.4 \times 10^{-6}$ | $-5.1 \times 10^{-4}$ *** | $-1 \times 10^{-3}$ *** | $-1.5 \times 10^{-3}$ *** | 26.1 *** |
| Assets | | $1.2 \times 10^{-5}$ | $5.3 \times 10^{-5}$ *** | $2.7 \times 10^{-5}$ | $-4 \times 10^{-4}$ *** | | $-2.6 \times 10^{-5}$ | $-2.6 \times 10^{-5}$ *** | $1 \times 10^{-5}$ | −14.1 *** |
| Extreme Value Distribution Predictor | | | | | | | | | | |
| Value-at-Risk 99 | $4.2 \times 10^{5}$ * | $-2.1 \times 10^{-7}$ | $-9.9 \times 10^{-7}$ | $-6 \times 10^{-6}$ | $1.4 \times 10^{-7}$ | 0.00 | $-7.4 \times 10^{-8}$ | $1.8 \times 10^{-6}$ *** | $-2 \times 10^{-9}$ | $5.8 \times 10^{-3}$ *** |
| N | 4850 | 8951 | 8372 | 8007 | 7601 | 12,335 | 7354 | 6299 | 7025 | 14,319 |
|  | Panel D | | | | | | | | | |
| **Independent Variable** | **2010** | **2011** | **2012** | **2013** | **2014** | **2015** | **2016** | **2017** | **2018** | **2019** |
| CAPM Predictor | | | | | | | | | | |
| Market Return | −0.06 | 48 *** | −37.8 *** | 23 | 25 *** | 0.46 | 0.61 | −0.41 | −1.36 | 7.17 |
| Beta | −0.05 ** | 1.7 *** | −1.9 *** | 0.25 | 0.01 | $-9.9 \times 10^{-5}$ | 0.02* | −0.02 * | −0.01 | |
| Fama–French Predictor | | | | | | | | | | |
| Market Capitalization | 0.003 | −0.04 *** | $-1.4 \times 10^{-2}$ *** | 0.80 | 0.05 *** | $-9.1 \times 10^{-6}$ | $5 \times 10^{-3}$ *** | $1.7 \times 10^{-3}$ | $-2.1 \times 10^{-4}$ | −0.06 |
| Book-to-Market | 3.07 | $-8.8 \times 10^{-5}$ | $4.3 \times 10^{-4}$ *** | $-2.1 \times 10^{-4}$ | $-6.2 \times 10^{-4}$ | 0.08 | $-1.5 \times 10^{-6}$ | $-6.5 \times 10^{-5}$ | $6.0 \times 10^{-5}$ | 11.38 |
| EBIT | $9.5 \times 10^{-5}$ | $-2 \times 10^{-4}$ | $-5.2 \times 10^{-5}$ | $-2.9 \times 10^{-3}$ | $8.4 \times 10^{-4}$ *** | $7.3 \times 10^{-6}$ | $-3.3 \times 10^{-4}$ *** | $-6.7 \times 10^{-4}$ *** | $-1.3 \times 10^{-3}$ *** | 16.3 *** |
| Assets | | $1.3 \times 10^{-5}$ | $-2.6 \times 10^{5}$*** | $-0.5 \times 10^{-5}$ | $-3.6 \times 10^{-4}$ | | $8.9 \times 10^{-6}$ | $6.0 \times 10^{-6}$ | $2.7 \times 10^{-5}$ | −10.2 *** |
| Extreme Value Distribution Predictor | | | | | | | | | | |
| Value-at-Risk 99 | $6.5 \times 10^{6}$ | $-3 \times 10^{-7}$ | $4 \times 10^{-8}$ | $-3 \times 10^{-6}$ | $-4 \times 10^{-8}$ | 0.00 | $3.4 \times 10^{-9}$ | $4.4 \times 10^{-8}$ | $1.9 \times 10^{-8}$ | $4.3 \times 10^{-3}$ *** |
| N | 4850 | 8951 | 8372 | 8007 | 7601 | 12,335 | 7354 | 6299 | 7025 | 14,319 |

**Table 5.** *Cont.*

| | | | | | Panel E | | | | | |
|---|---|---|---|---|---|---|---|---|---|---|
| **Independent Variable** | **2010** | **2011** | **2012** | **2013** | **2014** | **2015** | **2016** | **2017** | **2018** | **2019** |
| CAPM Predictor | | | | | | | | | | |
| Market Return | −0.23 | 50 *** | −45.3 *** | −28 | 25 *** | 0.44 | 1.09 | 0.27 | −1.1 | 1.77 |
| Beta | 0.59 *** | 3.6 *** | 1.0 *** | 0.61 | 0.04 | $8.5 \times 10^{-3}$ | 0.02 * | −0.01 | $1.3 \times 10^{-3}$ | −1.2 |
| Fama–French Factor | | | | | | | | | | |
| Market Capitalization | $-7 \times 10^{-3}$ | $-8.9 \times 10^{-3}$ | 0.01 ** | 0.74 | 0.11 *** | $1.6 \times 10^{-4}$ | $4.3 \times 10^{-3}$ ** | $-3 \times 10^{-3}$ ** | $-1.2 \times 10^{-3}$ | −0.73 |
| Book-to-Market | −1.33 | $-2.5 \times 10^{-4}$ | $5.4 \times 10^{-4}$ *** | $-1.1 \times 10^{-3}$ | $-9.6 \times 10^{-4}$ | 0.99 | $8.2 \times 10^{-5}$ | $-2.0 \times 10^{-4}$ | $-5.8 \times 10^{-4}$ | −80 |
| EBIT | $1.7 \times 10^{-4}$ | $-8.3 \times 10^{-4}$ | $1.1 \times 10^{-3}$ ** | −0.01 | $-1.1 \times 10^{3}$ *** | $5.7 \times 10^{-6}$ | $-1.8 \times 10^{-4}$ *** | $3.2 \times 10^{-4}$ *** | $4.8 \times 10^{-4}$*** | −3.5 *** |
| Assets | | $4.2 \times 10^{-5}$ ** | $-4.1 \times 10^{-5}$ *** | $6.8 \times 10^{-5}$ | $-2.3 \times 10^{-5}$ | | $5.2 \times 10^{-6}$ | $1.8 \times 10^{-5}$ ** | $4.8 \times 10^{-5}$ ** | 0.42 ** |
| Extreme Value Distribution Factor | | | | | | | | | | |
| Value-at-Risk 99 | $-2.6 \times 10^{-5}$ | $-4 \times 10^{-7}$ | $4.5 \times 10^{-7}$ | $1 \times 10^{-6}$ | $-5.2 \times 10^{-7}$ | 0.00 | $6.8 \times 10^{-8}$ | $-1.1 \times 10^{-6}$ ** | $3 \times 10^{-9}$ | $2.6 \times 10^{-3}$ |
| N | 4850 | 8951 | 8372 | 8007 | 7601 | 12,335 | 7354 | 6299 | 7025 | 14,319 |

* $p < 0.05$, ** $p < 0.01$, *** $p < 0.001$.

**Table 6.** Panel A. Predictors of daily excess returns with Expected Shortfall and Three Fama–French Factors. Panel B. Predictors of daily excess returns with Expected Shortfall and Five Fama–French Factors. Panel C. Predictors of daily excess returns with Expected Shortfall with 20% quantiles. Panel D. Predictors of daily excess returns with Expected Shortfall with 50% quantiles. Panel E. Predictors of daily excess returns with Expected Shortfall with 80% quantiles.

| Panel A | | | | | | | | | | |
|---|---|---|---|---|---|---|---|---|---|---|
| **Independent Variable** | **2010** | **2011** | **2012** | **2013** | **2014** | **2015** | **2016** | **2017** | **2018** | **2019** |
| CAPM Predictor | | | | | | | | | | |
| Market Return | 3.0 *** | 6.86 | 4.3 * | 1.54 ** | −8.7 | −3.6 | −0.93 | 3.18 | 5.77 | −4.7 |
| Beta | $3.3 \times 10^{-3}$ ** | 0.04 | 0.05 | $4.4 \times 10^{-3}$ | −0.04 | 0.18 | 0.03 | 0.04 | 0.08 | 0.09 |
| Fama–French Predictor | | | | | | | | | | |
| Market Capitalization | $-9.9 \times 10^{-3}$ * | $1.2 \times 10^{-3}$ | $2.4 \times 10^{-3}$ | $2.8 \times 10^{-3}$ | $7.9 \times 10^{-3}$ | −.02 | $6.7 \times 10^{-3}$ | $-5.9 \times 10^{-5}$ | $1.2 \times 10^{-3}$ | 0.07 * |
| Book-to-Market | −8.0 | $5.4 \times 10^{-5}$ | $-2.3 \times 10^{-5}$ | 0.00 | $-1.9 \times 10^{-5}$ | −2.4 | $-1.2 \times 10^{-5}$ | $5.1 \times 10^{-6}$ | $6.6 \times 10^{-7}$ | 18.7 |
| Extreme Value Distribution Predictor | | | | | | | | | | |
| Expected Shortfall | $-3.8 \times 10^{-7}$ | $-9 \times 10^{-8}$ | $-2.1 \times 10^{-6}$ * | 0.00 | $7.4 \times 10^{-8}$ | $-1.8 \times 10^{-6}$ *** | $7.1 \times 10^{-7}$ | $-4.6 \times 10^{-7}$ | $4.7 \times 10^{-8}$ | $-1.5 \times 10^{-6}$ *** |
| N | 16,940 | 8513 | 8055 | 7922 | 7074 | 16261 | 7011 | 6299 | 4779 | 7763 |
| $R^2$ | 0.05 | 0.04 | 0.03 | 0.08 | 0.03 | 0.02 | 0.05 | 0.17 | 0.03 | 0.03 |
| **Panel B** | | | | | | | | | | |
| **Independent Variable** | **2010** | **2011** | **2012** | **2013** | **2014** | **2015** | **2016** | **2017** | **2018** | **2019** |
| CAPM Predictor | | | | | | | | | | |
| Market Return | 1.3 *** | 6.96 | 4.3 * | 1.54 ** | −8.7 | −6.7 | −0.92 | 3.17 | 5.75 | −4.7 *** |
| Beta | −0.07 *** | 0.03 | 0.04 | $3.7 \times 10^{-3}$ | −0.04 | −0.02 | 0.04 | 0.03 | 0.05 | 0.09 |
| Fama–French Predictor | | | | | | | | | | |
| Market Capitalization | $3.6 \times 10^{-3}$ | $-1.9 \times 10^{-3}$ | $5.9 \times 10^{-3}$ | $4.1 \times 10^{-3}$ | $6.1 \times 10^{-3}$ | −0.02 | $4.6 \times 10^{-3}$ | $1.3 \times 10^{-3}$ | 0.02 | 0.08 * |
| Book-to-Market | −28 | $-4.2 \times 10^{-5}$ | $1.1 \times 10^{-5}$ | $2.2 \times 10^{-5}$ | $2.9 \times 10^{-3}$ | −28 | $6.3 \times 10^{-4}$ | $1.3 \times 10^{-4}$ | $4.7 \times 10^{-4}$ | 19.1 |
| EBIT | $-3.3 \times 10^{-5}$ | $1.4 \times 10^{-4}$ | $-1.7 \times 10^{-5}$ | $1.6 \times 10^{-5}$ | $-5.9 \times 10^{-5}$ | $2.7 \times 10^{-5}$ | $4.3 \times 10^{-6}$ | $-5.2 \times 10^{-6}$ | $-1.2 \times 10^{-4}$ | $1 \times 10^{-5}$ |
| Assets | | $5.8 \times 10^{-6}$ | $-3.7 \times 10^{-6}$ | $-1.08 \times 10^{-6}$ | $-1.2 \times 10^{-4}$ | | $-2.6 \times 10^{-5}$ | $-5.3 \times 10^{-6}$ | $-1.9 \times 10^{-5}$ | $-1.7 \times 10^{-5}$ |
| Extreme Value Distribution Predictor | | | | | | | | | | |
| Expected Shortfall | $1 \times 10^{-8}$ | $5.9 \times 10^{-8}$ | $-2.3 \times 10^{-6}$ * | $3.1 \times 10^{-8}$ | $1.1 \times 10^{-7}$ | $5 \times 10^{-6}$ *** | $5.9 \times 10^{-7}$ | $-4.4 \times 10^{-7}$ | $7.3 \times 10^{-8}$ | $-1.1 \times 10^{-6}$ *** |
| N | 4855 | 8451 | 8055 | 7922 | 074 | 10,776 | 7011 | 6299 | 4779 | 7763 |
| $R^2$ | 0.05 | 0.04 | 0.03 | 0.08 | 0.03 | 0.03 | 0.05 | 0.17 | 0.03 | 0.03 |

**Table 6.** *Cont.*

| Independent Variable | 2010 | 2011 | 2012 | 2013 | 2014 | 2015 | 2016 | 2017 | 2018 | 2019 |
|---|---|---|---|---|---|---|---|---|---|---|
| **Panel C** | | | | | | | | | | |
| CAPM Predictor | | | | | | | | | | |
| Market Return | 0.05 | 42 *** | −40 *** | −9.2 | 0.73 | 0.46 | −16.9 | 0.67 | −26 | 6.93 |
| Beta | −0.5*** | −0.88 *** | −0.35 *** | −11 *** | −0.07 * | $−9 \times 10^{-3}$ | −28 *** | −0.06 *** | −5.27 | 0.35 |
| Fama–French Predictor | | | | | | | | | | |
| Market Capitalization | $−4.4 \times 10^{-4}$ | −0.05 *** | 0.01 ** | −0.1 *** | $1 \times 10^{-3}$ | $−2.4 \times 10^{-4}$ | −0.05 * | $3.7 \times 10^{-3}$ ** | 0.91 | 0.07 |
| Book-to-Market | −27 | $−1.4 \times 10^{-4}$ | $−7.8 \times 10^{-4}$*** | $1.9 \times 10^{-4}$ | $5 \times 10^{-4}$ | −0.77 | 0.06 *** | $−1.2 \times 10^{-4}$ | −1.8 *** | 14.24 |
| EBIT | $1 \times 10^{-3}$ *** | $4.3 \times 10^{-4}$ | $1.5 \times 10^{-3}$ ** | $−1.3 \times 10^{-3}$ *** | $8 \times 10^{-5}$ | $−5.6 \times 10^{-6}$ | $4.7 \times 10^{-4}$ *** | $7.8 \times 10^{-4}$ *** | −1.8 *** | 5.65 *** |
| Assets | | $2.9 \times 10^{-5}$ | $3.6 \times 10^{-5}$ *** | $8.1 \times 10^{-6}$ | $−4 \times 10^{-6}$ | | $−2.5 \times 10^{-3}$ *** | $−8.2 \times 10^{-7}$ | 0.06 *** | 6.04 *** |
| Extreme Value Distribution Predictor | | | | | | | | | | |
| Expected Shortfall | $9 \times 10^{-8}$ | $2.1 \times 10^{-8}$ | $1.5 \times 10^{-6}$ *** | $−1.6 \times 10^{-6}$ *** | $4.3 \times 10^{-7}$ *** | 0.00 | $−3.2 \times 10^{-6}$ | $6.1 \times 10^{-7}$ *** | $2.9 \times 10^{4}$ *** | $2.1 \times 10^{-4}$ |
| N | 4855 | 8451 | 8055 | 7922 | 7074 | 10,776 | 7011 | 6299 | 4779 | 7763 |
| **Panel D** | | | | | | | | | | |
| CAPM Predictor | | | | | | | | | | |
| Market Return | −0.13 | 48 *** | −23 *** | −9.2 | 0.33 | 0.44 | −25 | 0.11 | −8.77 | 8.61 |
| Beta | 2.8 *** | −0.37 *** | 0.85 *** | −3.7 *** | 0.19 *** | $−9.3 \times 10^{-5}$ | −14 *** | −0.02 * | −1.14 | 0.07 |
| Fama–French Factor | | | | | | | | | | |
| Market Capitalization | $−2.6 \times 10^{-3}$ | −0.04 *** | 0.04 *** | −0.05 *** | $−7.8 \times 10^{-4}$ | $−8.07 \times 10^{-6}$ | −0.06 ** | $2.2 \times 10^{-3}$ | 0.26 | −0.45 |
| Book-to-Market | −67 | $2.6 \times 10^{-5}$ | $3.9 \times 10^{-5}$ | $−2.7 \times 10^{-4}$ * | $3.6 \times 10^{-6}$ | 0.07 | 0.03 ** | $2.4 \times 10^{-4}$ | −2.6 *** | 15.56 |
| EBIT | $−3.4 \times 10^{-4}$ *** | $−7.5 \times 10^{-4}$ | $−1.5 \times 10^{-4}$ | $−1 \times 10^{-3}$ *** | $−1.2 \times 10^{-5}$ | $7.4 \times 10^{-7}$ | 0.001 *** | $5.4 \times 10^{-4}$ *** | −0.05 | 0.30 |
| Assets | | $7.6 \times 10^{-6}$ | $4.3 \times 10^{-5}$ *** | $3 \times 10^{-5}$ *** | $1.2 \times 10^{5}$ | | $−1.3 \times 10^{-3}$ ** | $−1.4 \times 10^{-5}$ ** | 0.11 *** | 11.0 *** |
| Extreme Value Distribution Factor | | | | | | | | | | |
| Expected Shortfall | $1.9 \times 10^{-7}$ | $−1.4 \times 10^{-8}$ | $8.5 \times 10^{-7}$ ** | $−1.7 \times 10^{-8}$ | $−1 \times 10^{-8}$ | 0.00 | $−3.5 \times 10^{-5}$ *** | $6 \times 10^{-8}$ | $3 \times 10^{-5}$ | 0.00 |
| N | 4855 | 8451 | 8055 | 7922 | 7074 | 10,776 | 7011 | 6299 | 4779 | 7763 |

**Table 6.** *Cont.*

| | | | | | Panel E | | | | | |
| --- | --- | --- | --- | --- | --- | --- | --- | --- | --- | --- |
| Independent Variable | 2010 | 2011 | 2012 | 2013 | 2014 | 2015 | 2016 | 2017 | 2018 | 2019 |
| CAPM Factor | | | | | | | | | | |
| Market Return | 0.7 | 56 *** | −40.5 *** | −9.2 | −0.66 | 0.41 | 21 | −0.86 | −0.02 | 6.97 |
| Beta | 3.2 *** | 0.92 *** | 3.5 *** | −0.34 *** | 0.18 *** | $7.3 \times 10^{-3}$ | −11 *** | $-1.4 \times 10^{-3}$ | 6.84 | 4.79 |
| Fama–French Factor | | | | | | | | | | |
| Market Capitalization | −0.01 * | $3.9 \times 10^{-3}$ *** | 0.05 *** | −0.04 *** | $-1.8 \times 10^{-3}$ | $4.1 \times 10^{-4}$ | −0.03 | $-2 \times 10^{-3}$ | −0.33 | −2.46 |
| Book-to-Market | 20 | $2.9 \times 10^{-4}$ | $-3.3 \times 10^{-4}$ ** | $-5.7 \times 10^{-4}$ *** | $-4.3 \times 10^{-4}$ | 0.81 | $-3.4 \times 10^{-3}$ | $-3.3 \times 10^{-4}$ * | −1.72 *** | 67.82 |
| EBIT | $-2.7 \times 10^{-4}$ * | $-1.2 \times 10^{-3}$ ** | $1.1 \times 10^{-3}$ *** | $-1 \times 10^{-3}$ *** | $-6.5 \times 10^{-5}$ | $5.2 \times 10^{-6}$ | $1.1 \times 10^{-4}$ | $4.9 \times 10^{-3}$ *** | 2.59 *** | 0.18 |
| Assets | | $-2.4 \times 10^{-5}$ | $-9.8 \times 10^{-5}$ *** | $4.6 \times 10^{-5}$ *** | $1.8 \times 10^{-5}$ | | $1.2 \times 10^{-4}$ | $1.8 \times 10^{-5}$ | 0.06 *** | 17.6 *** |
| Expected Value Distribution Factor | | | | | | | | | | |
| Expected Shortfall | $-1.5 \times 10^{-7}$ | $-4.6 \times 10^{-8}$ * | $7.6 \times 10^{-8}$ | $1.1 \times 10^{-6}$ ** | $-6.5 \times 10^{-8}$ | 0.00 | $-3.2 \times 10^{-6}$ | $3.9 \times 10^{-7}$ | $-1.3 \times 10^{-5}$ | $-4.4 \times 10^{-4}$ |
| N | 4855 | 8451 | 8055 | 7922 | 7074 | 10776 | 7011 | 6299 | 4779 | 7763 |

* $p < 0.05$, ** $p < 0.01$, *** $p < 0.001$.

## 5. Conclusions

### 5.1. Discussion of Results

Existing literature sets forth that it is beneficial to be green. Returns on green securities were shown to be higher mainly due to a positive image from media attention (Clarkson et al. 2011; Hamilton 1995). Hart and Ahuja (1996) found that measures to reduce carbon emissions were directly related to positive financial performance. Konrad (2010) created a green security trading strategy that outperformed market returns. Using a more contemporary sample from 2002 to 2010, Levi and Newton (2016) disentangled media attention from Fama–French predictors and risk measures. Their comparison of green and nongreen securities found a 3.7% return advantage to green investments in the long-run, though this benefit dissipates in the short-run. Risk did not provide any explanation of long-run returns. Yet, in other studies, green investors demanded excess returns to compensate for additional risk, to the tune of 7–18 basis points in the Goss and Roberts (2011), and some additional return in the Bouslah et al. (2013) study. We conjecture that extreme value risk in the form of Value-at-Risk influences returns for ultra-short holding periods, of a single year. The Levi and Newton (2016) finding of no risk effects on returns may be due to a longer time horizon. There may be some explanation of returns by Fama–French factors, as was found by Levi and Newton (2016) as well. However, the Fama–French factors were really control variables, with Value-at-Risk explaining most of the excess return in green securities.

Our results advance knowledge of the predictors of excess returns in green energy securities in four ways. First, we consider the market model. Market returns explained excess returns significantly, particularly about 2010, or in the early years. At their inception, green energy securities had co-movements similar to a portfolio of a major index of the broad market, such as the S & P 500 index. This result is in keeping with the early studies of the market model explaining the variation in green energy returns (Lesser et al. 2014), and the increase in excess returns due to increases in the market return. Green energy securities of that era were similar to the large cap stocks of the broad market. The introduction of small cap stocks, as noted by Ibikunle and Steffen (2017), ushered the introduction of tail risk, a result we document in the 2017–2019 years. Second, we note that systematic risk, measured by beta in the market model, has little effect on the excess returns of green energy securities. Any effect of beta on excess returns is due to firms of differential size, differential book-to-market equity, differential levels of fixed investment, or different profitability in the sample. Once the Fama–French Five-Factor model accounted for these variations, beta failed to display any significant impact on excess returns. Market risk, based on covariance between green stock prices and the excess return of the broad market and the risk-free rate, did not affect excess returns. Third, a conditional extreme value distribution may provide a closer approximation of the distribution of green energy portfolio values than a traditional normal distribution. The powerful explanatory power of the two Value-at-Risk measures suggests that extreme values up to the Value-at-Risk percentiles significantly reduced excess returns, particularly for extreme values of excess returns, such as at the 20% quantile, and the 80% quantile. Finally, Expected Shortfall adds to the informativeness of the Value-at-Risk measure in explaining excess returns. Value-at-Risk is the lower bound of Expected Shortfall. While Value-at-Risk indicates the percentile up to which losses affect excess returns, it does not define losses beyond that percentile. Figure 1 provides a graphic depiction of the relationship between Value-at-Risk and Expected Shortfall in the left tail of a leptokurtic extreme value distribution. Expected Shortfall obtains the average of the losses beyond the Value-at-Risk percentile, resulting in its ability to measure the losses at extremely low values of excess returns, that influence excess returns. In 2012,2013,2014, and 2017, and 2018, the lowest 20% of excess returns are explained by the average losses defined by the Expected Shortfall. We may combine this result with the literature. The extreme Value-at-Risk values in the tails that reduced excess returns in our study parallel the extreme downward movements of green energy indices due to tail risk, observed in the Bouri et al. (2017) study. Our study used green energy securities, while their study used green energy indices, so that the similarity of results suggests that tail risk exists for

green energy securities, regardless of whether indices or securities are used. Perhaps the extreme returns in the tails are explained by crude oil prices, as Saeed et al. (2021) observed a high degree of connectedness between clean energy and fossil fuel energy returns in the tails. Future research should undertake a replication of the Saeed et al. (2021) study with our Value-at-Risk measures and oil prices to determine if their joint effect will improve the explanation of the variation in excess security returns. Sasodia et al. (2016) observed that regulation could have a negative effect on investment. Therefore, regulatory effects must be included as a control variable in our examination of the combined influence of crude oil prices and Value-at-Risk measures of green energy securities. Sasodia et al. (2016) other drivers of green energy investment may also be included as control variables, such as levelized cost, carbon emissions, and climatic conditions. As in the Sasodia et al. (2016) study, sample sizes may be varied to determine if the explanation of excess returns is robust to sample size differentials. If sample size influences results, results may be reported separately for each sample size.

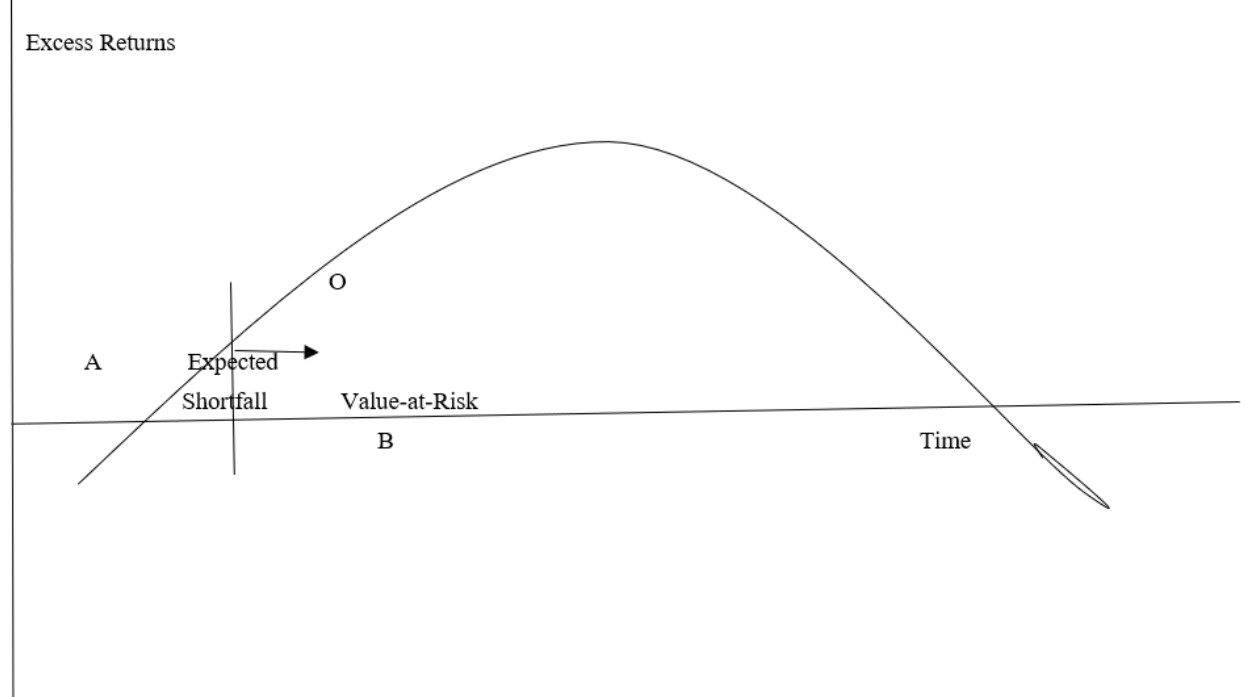

**Figure 1.** Line *OB* is the Value-at-Risk percentile, which is the upper bound of the Expected Shortfall, which is contained in area *AOB*, in the lower tail of the extreme value distribution.

A noteworthy study is that of Liu et al. (2021), who measured the degree of market integration across green energy investments, fossil fuel investments, and market indices. This strand of literature is valuable for investors seeking to combine different investments, under varying market conditions. They found that the driving factors of market integration were uncertainty measures at high and low levels of market integration. VIX (measure of risk in the US stock market), OVX (measure of uncertainty in the crude oil market), EPU (a daily frequency uncertainty metric), FSI (financial stress index), and EMV (equity-market volatility tracker) were drivers at high levels of market integration, while VIX, EPU, and FSI were significant when mean-based estimators were used. Our results indicate that uncertainty in green energy investments may be measured by extreme values at the Value-at-Risk and Expected Shortfall levels in the tails. Both Value-at-Risk and Expected Shortfall may act as uncertainty measures that drive high levels of market integration. They are closest by definition to EMV, the equity market volatility tracker, as our study was restricted to equities. As EMV drives market integration, Value-at-Risk and Expected Shortfall may integrate green energy equities, fossil fuel investments, and market indices. For portfolio

managers, these portfolios may be secure investments, during periods of high uncertainty, such as the COVID-19 crisis. Diversification of risk may be accomplished by such highly integrated market portfolios, as may rational asset pricing. By combining such diverse investments, during periods of uncertainty, risk is not only reduced, but assets become more rationally priced, as irrational expectations are diversified away with the addition of stable market indices to less stable commodity investments.

Hence, we view Value-at-Risk and Expected Shortfall as supporting market integration.

This result follows past research which viewed market integration as the outcome of combining stocks, bonds, and commodities (see Liu et al. 2021, for a review).

### 5.2. Limitations of This Study and Scope for Future Research

This study employed factor models, in which risk measures were not built using a buy–sell approach. Future research should overcome this limitation by creating risk measures, using a buy–sell approach.

This study examined the impact on excess returns of predictors based on market return and beta from CAPM. It did not examine CAPM anomalies. Given the strong significance of market returns in explaining excess returns, CAPM anomalies should be examined for their effects on excess returns. Abnormal returns prior to dividend announcements, merger announcements, and earnings announcements are expected (see Mitchell et al. 2004, for a review. Do green energy securities have such abnormal returns prior to events? Can abnormal returns be explained by market returns, or are there other predictors? Future research should be directed to these questions. Mitchell et al. (2004) found that short interest predicted abnormal returns, so that other predictors may explain excess returns in green energy securities.

We limited our examination of returns to an ultra-short time horizon, i.e., to investors who buy and hold for a single year. As results may differ for longer holding periods, future research should consider intermediate-term and long-term holding periods.

This study employed a panel regression, followed by a quantile regression. The quantile regression results reinforce the panel data regressions for extreme values of excess returns in the 20th and 80th quantiles. Yet, as Atsalakis et al. (2021) showed, a quantile-on-quantile regression may yield different results in other quantiles. Therefore, this study is limited by its omission of a quantile-on-quantile regression, which may yield other predictors of excess returns, than those found. Future research should perform such a quantile-on-quantile regression.

As green energy production is current and topical, abnormal returns are likely to precede the formation of green energy startups. Diether et al. (2002) tested the Miller (1977) model, verifying that irrational price optimism, in which optimistic investors purchase securities, but rational investors do not sell due to high short-sale costs. Future research should uncover irrational optimism in green energy securities, and determine if only irrational optimists trade in a test of the Miller (1977) model.

As noted, this study did not examine the stationarity of variables, as it was confined to an examination of ultra-short returns. Future research may stipulate that all variables be stationary in order to establish that stable long-term relationships exist among variables, with less stable short-term relationships for intermediate-term and long-term holding periods.

### 5.3. Implications for Investors

What are the implications of these results for investors? In the early years of this study, i.e., about 2010–2011, investors would have found that a portfolio of green equities closely followed the market, with returns similar to the S & P 500. Subsequently, the portfolio would have included stocks of smaller, less stable firms, resulting in extreme value returns, represented by significant Value-at-Risk percentiles. To minimize the probability of significantly low excess returns, investors would need to invest in portfolios in which the risk of loss was at the Value-at-Risk percentiles, as these were known extreme values of excess returns with a 1-5% risk of losses. It is recommended that investors limit their exposure

to risk by refraining from pursuing the extremely low returns in the tails, measured by Expected Shortfall, as this strategy would expose them to excessively uncertain risk, beyond the known risk represented by the Value-at-Risk percentiles. Why would investors pursue extremely low excess returns? The response to this query is that such returns could form part of a portfolio immunization strategy, designed to balance a high-return/high-risk portfolio. In other words, if investors have high-return portfolios with considerable risk, they may seek to reduce risk by diversifying with green securities, reasoning that low returns in some of the green securities in the tails could have low risk.

### 5.4. Policy Implications

Sasodia et al. (2016) identified regulatory perception as having a negative impact on renewable energy investments in the European Union, primarily due to the high cost of initial installation. This negative perception could stimulate the perception of high risk of renewable energy securities. Our findings support this perception, with extreme value risk measures, such as Value-at-Risk in the 95th and 99th percentiles, explaining the most variation in excess returns of green energy securities. Intuitively, subsidies for green energy installation reduce cost perceptions, resulting in greater consumer and producer interest in investing in green energy. If subsidies decline, consumers will be disincentivized to install solar power, or purchase electric vehicles. This would lead to increasing risk perceptions for green energy securities, as supported by our finding of the explanation of excess returns by extreme value distributions, rather than market models. Regulators may respond by offering incentives to reduce the risk of energy installation, thereby reducing the negative perception of renewable energy investments. Such incentives, such as tax breaks, must be made permanent, as their expiration will increase the perception of negative regulatory perception of green investments. Policy makers must seek public–private partnerships, whereby a combination of tax incentives to individuals, and government subsidies to installers of solar energy, or owners of wind farms would further reduce the cost of installation, in order to increase public support for renewable energy.

As Sasodia et al. (2016) found, reduction in carbon emissions significantly predicted investment in renewable energy by countries in the European Union. This is expected, given that reduction in carbon emissions diminishes the risk of climate change. Regulatory frameworks that reduce carbon emissions would increase the returns of green energy securities (Saltari and Travaglini 2011). Our findings suggest that market returns would explain these excess returns, as regulatory action to incentivize renewables production would apply uniformly to large and small green energy producers, making them all similar to each other. The uncertainties of small cap green energy stocks would be eliminated, so that they could perform similarly to large cap stocks, or a market index. A reduction of uncertainty in small cap green energy investments may significantly increase their attractiveness to portfolio managers. Small cap green energy investments may be recommended for aggressive growth portfolios, or added to green energy index funds, with the expectation of high returns, with reduced risk. The reduced risk may make small cap green energy securities a choice for retirement savings, as they may contribute to long-term portfolio growth. Such high returns with reduced risk may appeal to investors who are both socially responsible and thrill-seeking. Such investors are committed to investments that protect the environment, while seeking excitement in their investments. Small firms in renewable energy easily meet both of these requirements of socially responsible investing, and being novel enough to add excitement to portfolio creation.

**Author Contributions:** Conceptualization—R.A. and Z.T.; methodology—R.A.; software—R.A.; validation—R.A. and Z.T.; formal analysis—R.A. and H.E.-C.; investigation—R.A. and H.E.-C.; resources—Z.T.; data curation—R.A.; writing—original draft preparation—R.A. and Z.T.; writing—review and editing—H.E.-C.; visualization—R.A.; supervision—H.E.-C. and Z.T.; project administration—R.A. All authors have read and agreed to the published version of the manuscript.

**Funding:** This research received no external funding.

**Data Availability Statement:** Data are available from Abraham upon request.

**Conflicts of Interest:** The authors declare no conflict of interest.

**Appendix A**

List of Stocks in the Sample

1. Atlantic Yield Plc
2. Advanced Disposal Services
3. American Electric Power
4. Airgain, inc
5. Air Products & Chemicals
6. Ampliphi Biosciences Corp.
7. Armata Pharmaceuticals Inc.
8. Atlantic Power Corp.
9. Avalon Holdings Corp.
10. Atlantic yield Plc.
11. Brookfield Renewable Partners
12. Benchmark Electronics
13. Constellium NV
14. Clean Harbors, Inc.
15. Covanta holdings Corp.
16. US Ecology
17. Fidelity Merrimack Street Trust
18. NRC Group Holdings Corp.
19. Pattern Energy Group, Inc.
20. Perma Fix Environmental Services, Inc.
21. Servicemaster Global Holdings
22. Sky Solar Holdings Ltd.
23. Sharps Compliance Corp.
24. Stericycle Inc
25. Tecogen Inc
26. York Water Co.
27. Aim Immunotech Inc.
28. Cincinnati Bell Inc.
29. Mr Cooper Group
30. DTE Energy
31. Estaline Technologies
32. Forum Energies Technologies
33. Fluor Corp New
34. Hemispherx Biopharma
35. Johnson Controls International Plc
36. Nextera Energy
37. Ormat Technologies
38. Plug Power
39. Renewable Energy Group
40. Suncor Energy Inc New
41. Waste Management
42. Advanced Energy
43. Bloom Energy
44. Canadian Solar
45. Cheniere Energy
46. Clearway Energy
47. Daqo New Energy
48. Enphase Energy
49. Fuelcell Energy

50. Enviva Partners LP
51. Hannon Armstrong
52. Jinkosolar
53. Nexterra
54. Ocean Power Technologies
55. Renesola
56. Renewable Energy Group
57. Sunpower Corp.
58. Sunrun
59. TPI Composites
60. BWX Technologies
61. General Electric Inc.
62. Solaredge Technologies Inc.
63. Azure Power Global Limited
64. General America Investors, Inc.
65. Sunnova Energy International
66. Standex International Corp.
67. Neonode, Inc.
68. American Super Conductor Corp.
69. India Fund, Inc.
70. Ballard Power Systems
71. Arca Biopharma
72. Asbury Automotive Group
73. Proshares Trust
74. First Trust Global
75. Ishares Trust
76. Tesla Inc.
77. Ameresco Inc
78. Gevo Inc.
79. Atlantic yield Plc.
80. avangrid, Inc.
81. Covanta Holding Corp.
82. Green Plains, Inc.
83. PG&E Corp.
84. Alphabet Inc
85. Consol Coal Resources LP
86. Edison International
87. Mosaic Acquisition Corp De
88. Southern Co.
89. Exelon Corp.
90. Idacorp Inc.
91. ppl Corp.
92. Firstenergy Corp.
93. CM Energy Corp.
94. WEC Energy Group Inc.
95. XCEL Energy Inc.
96. Entergy Corp New
97. EQT Corp.
98. Avista Corp.
99. Ameren Corp.
100. Duke Energy Corp, New.
101. Eversource Energy
102. Invacare Corp.
103. Otter Tail Corp.

104. Dominion Energy
105. Rex American Resources Corp.
106. American States Water Corp.
107. Brookfield Asset Management
108. Gladstone Commercial Corp.
109. Portland GE
110. Halcon Resources Corp.
111. Torchlight Energy Resources
112. Vivint Solar
113. Tantech Holdings
114. SPT Energy Co.
115. Vivopower International
116. Trinity Industries
117. Dreyfus Strategic Municipal Bond Fund A
118. BNY Mellon Strategic Municipal Bond Fund
119. Fueltech
120. Companhia Paranaense de Energia
121. Clean Energy Funds Corp.
122. Companhia Energetica De Minas GE

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
