# Peer review of "Predictors of Excess Return in a Green Energy Equity Portfolio: Market Risk, Market Return, Value-at-Risk and or Expected Shortfall?"

_jrfm, doi:10.3390/jrfm15020080_

Round 1

Reviewer 1 Report

This paper considers the predictors of excess equity returns in a portfolio of global green energy producers using data covering the period 2010-2019. The methods applied are panel and quantile regressions. The results indicate the importance of the extreme value theory to explain the excess returns of green energy equity portfolio.

The research question is relevant and has implications for many market participants in green energy investments. My comments are as given below:

I- I would have liked to see how the empirical analysis and the key empirical findings presented in this paper are related to the literature on the green investment markets and its integration with conventional stock markets such as : Liu, X., Bouri, E., and Jalkh, N. (2021). Dynamics and determinants of market integration of green, clean, dirty energy investments and conventional stock indices. Frontiers in Environmental Science, 9, https://doi.org/10.3389/fenvs.2021.786528 . This would add more value to your findings and improve their relevance to the related literature. This is useful given that previous studies have considered the drivers of return of green energy investments and the market integration between.

2- Extend the economic tuition of the results and discuss them in light of previous studies in a thorough way. While doing so, consider the differences in the results of the factor models as compared to the literature on conventional stock markets, which can give insights on the particularity of the green investments as compared to non-green.

3-This is unclear: “ Yet, in the left tail, low risk may or may not be found, as risk beyond the Value-at-Risk levels is unknown.”.

4-         Try to revise the conclusion section in order to give more detailed policy for the sake of green investors, portfolio mangers, and policymakers. What are the limitations of your study.

5-The format of tables can be improved and the tables should be self-explanatory.

Author Response

See page 34 of the attached document. Changes are in yellow. 

Reviewer 1: (See changes in yellow)

1.Relate empirical findings to literature on the integration of results with conventional stock markets.

See Page 26, Section 5.1. The last paragraph describes the Liu et al. (2021) study, and others on market integration.

  1. Discuss results thoroughly with reference to literature on factor models, and green and non green investments. See page 24, paragraph 1. Section 5.1, Discussion of Results. Multiple studies have been cited to integrate our findings into the literature.
  2. There is an unclear sentence in Section 5.3, Page 27. The green line eliminates this sentence.
  3. Provide policies in Conclusions and limitations of the study. See pages 27-28, Section 5.4. Policies for investors, regulators, and portfolio managers have been added. Limitations are in Section 5.2, on pages 26-27.
  4. Tables should be self-explanatory. Table titles have been shortened, and made more descriptive on pages 7, 8, 11, 12, 13, 14, 15, 16, 17, 18, 19, 20, 21, 22, 23, 24.

Reviewer 2 Report

I believe that the authors did a good job in the review, after the rejection. I just think that they should revise the way they present the results, once the tables are unreadable. Appendix A also needs title.

Author Response

See attached document, items in gray, and tables (titles changed). 

Reviewer 2: (See changes in gray)

1.Tables are unclear. Table titles have been shortened, and made more descriptive on pages 7, 8, 11, 12, 13, 14, 15, 16, 17, 18, 19, 20, 21, 22, 23, 24.

2.Appendix A has a title, on page 30.

Round 2

Reviewer 1 Report

I am satisfied with the revised manuscript. 

This manuscript is a resubmission of an earlier submission. The following is a list of the peer review reports and author responses from that submission.

Round 1

Reviewer 1 Report

JRFM: Predictors of Excess Return in a Green Energy Equity Portfolio: Market Risk, Market Return, Value-at-Risk and or Expected Shortfall?

The authors of this paper examine the predictors of excess equity returns in a portfolio of global green energy producers over the sample period 2010-2019. To this end, they use panel data and quantile regressions and the reported results show the importance of the extreme value theory to explain the excess returns of green energy equity portfolio.

The research topic is appealing and important to many market participants in light of the growing interest in green energy investments. The employed panel methods are suitable. I provide several feedbacks on the positioning of the paper given the rising literature on green energy investments, and their relationship with other assets and the factors that drive them, and the added value of the findings. Once, these are addressed, I will reconsider my decision. My comments are as follows:

1- In the introduction section, the authors should better present the added value and their genuine contribution in light of previous studies.

2- The data section should more clearly define and describe the dataset and names of clean energy equity investments used in the analysis, besides the construction of the portfolio.  Furthermore, provide further motivation regarding the choice of the sample period.

3- I would have liked to see how the empirical analysis and the key empirical findings presented in this paper are related to findings from previous papers. This would add more value to the analyses and make it more susceptible to citations from the rising literature on green energy investments. Accordingly, when discussing your results you should take into account that your paper is related to a strands of literature concerning returns and connectedness between green energy investments and non-green energy investments. In this regard, you should consider the following papers: “Gold and crude oil as safe-haven assets for clean energy stock indices: Blended copulas approach. Energy, Vol. 178, pp. 544-553”; “Extreme return connectedness and its determinants between clean/green and dirty energy. Energy Economics, Vol. 96, 105017”; “The effect of sample size on European Union’s renewable energy investment drivers. Applied Economics, 48(53), 5129-5137.” You should use these studies to discuss your findings and put them in the context of the existing literature, especially given that some of these studies among other things consider the drivers of return of green energy investments. .

4- Importantly, try to add more economic tuition of the results. In other words, the results section requires the move beyond statistical significance . Furthermore, consider the discussion of results in light of the existing literature, including the papers suggested in my above comment.

5-Please give more explanation regrading regarding your suggestion about the following: “It is recommended that investors limit their exposure to risk by refraining from pursuing the extremely low returns in the tails, measured by Expected Shortfall, as this strategy would expose them to excessively uncertain risk, beyond the known risk represented by the Value-at-Risk percentiles.”.

6-         In the conclusion section, a more detailed discussion of the policy implications would make the paper richer and more informative for green investors, investors, and policymakers. Also, consider describing the study limitations and scope for future research. For example, given that you use panel and then quantile regression, it would be interesting to indicate that a possible extension, is to use the quantile on quantile panel regression as in “Natural disasters and economic growth: A quantile on quantile approach. Annals of Operations Research, Vol. 306 No. 1-2, pp. 83–109”.

7-The format of tables should be improved as it is not acceptable in its current form. Importantly, make your tables more self-explanatory.

8- Make sure that all that all cited papers are presented in the reference list and vice-versa.

Reviewer 2 Report

In this paper, the authors propose to analyse the predictors of excess equity returns in a portfolio of global green energy producers, based on a dataset between 2010 and 2019, trhouth the use of fixed effects panel data regressions. Despite the potential interest of the paper, I believe that the paper should consider several improvements, such as:

1) The paper needs a deep literature review about the methods which are used. For example, when we look at section 2.1, we see that authors refer to some of the prior methods but it should be made a literature review about the use of CAPM, with same or different methods, which should be clearly discussed. We could find many and many applications about CAPM in the literature (should it being equally considered in the remaining theoretical sections). For example, for CAPM you could see https://www.mdpi.com/2673-8392/1/3/70/htm which presents several methods. 

2) The use of a time sample of 2010-2019 should be justified. Note that for the theme under analysis it would be relevant to use a wider time sample which I believe it could have more interesting results.

3) The way as tables are presented makes it very difficult to read the different results. Moreover, different tables have also different font sizes. Issues related with the presentation are very relevant in scientific works. See also the way as Figure 1 is presented.

4) Issues like non-stationarity were considered in the analysis? (I believe not)

5) The way as results are presented is also confusing. See, for example, the first paragraph of section 5.2.

6) It is also needed to make a linkage between the results and the theoretical issues.

In summary, despite the potential interest of the paper, I believe that it is not in conditions to continue the review process.

Reviewer 3 Report

The paper tries to propose an approach to evaluate the financial performance of green securities using risk indicators in a factor model.

There are many problems in this paper, of which some relevant ones are:

  • in the factor models the risk measures are not basically factors since they are not build using the typical buy-sell approach.
  • also endogeneity is likely to be a main issue, which is not discussed in the paper
  • the analysis is performed in the interval 2010-2019, thus not including recent periods which instead are more relevant for the topic (as also raised by Authors)
  • the review of the literature, and how the paper aims to contribute, is not clear and omits to consider very relevant papers appeared in the last few years
  • data selection: it is obscure how Authors selected the 122 firms and which type of bias they may have
  • with few firm observations (122) it is questionable the use of quantile regression. Also is it a cross-section or panel quantile regression?
  • Main analysis: Authors analyze Fixed Effects Panel Data Regressions and then report estimates for each year. It is not clear the econometric setup.